

# Symmetry decomposition of negativity of massless free fermions

Sara Murciano[1], Riccarda Bonsignori[1] and Pasquale Calabrese[1,2]

**1** SISSA and INFN Sezione di Trieste, via Bonomea 265, 34136 Trieste, Italy
**2** International Centre for Theoretical Physics (ICTP), Strada Costiera 11, 34151 Trieste, Italy

## Abstract

We consider the problem of symmetry decomposition of the entanglement negativity in free fermionic systems. Rather than performing the standard partial transpose, we use the partial time-reversal transformation which naturally encodes the fermionic statistics. The negativity admits a resolution in terms of the charge imbalance between the two subsystems. We introduce a normalised version of the imbalance resolved negativity which has the advantage to be an entanglement proxy for each symmetry sector, but may diverge in the limit of pure states for some sectors. Our main focus is then the resolution of the negativity for a free Dirac field at finite temperature and size. We consider both bipartite and tripartite geometries and exploit conformal field theory to derive universal results for the charge imbalance resolved negativity. To this end, we use a geometrical construction in terms of an Aharonov-Bohm-like flux inserted in the Riemann surface defining the entanglement. We interestingly find that the entanglement negativity is always equally distributed among the different imbalance sectors at leading order. Our analytical findings are tested against exact numerical calculations for free fermions on a lattice.


## 1 Introduction

The Rényi entanglement entropies are the most successful way to characterise the bipartite entanglement of a subsystem $A$ in a pure state of a many-body quantum system [1–4], also from the experimental perspective [5–9]. Given the reduced density matrix (RDM) $\rho_A$ of a subsystem $A$, obtained after tracing out the rest of the system $B$ as $\rho_A \equiv \mathrm{Tr}\rho$, the Rényi entropies are defined as

$$S_n = \frac{1}{1-n} \log \mathrm{Tr}\rho_A^n. \tag{1}$$

From these, the von Neumann entropy is obtained as the limit $n \to 1$ of Eq. (1) and also the entire spectrum of $\rho_A$ can be reconstructed [10]. The essence of the replica trick is that for integer $n$, in the path-integral formalism, $\mathrm{Tr}\rho_A^n$ is the partition function on an $n$-sheeted Riemann surface $\mathcal{R}_n$ obtained by joining cyclically the $n$ sheets along the region $A$ [11, 12]. Furthermore with the experimental settings developed so far [5–9], only Rényi entropies with integer $n$ are accessible.

For a mixed state, the entanglement entropies are no longer good measures of entanglement since they mix quantum and classical correlations (e.g. in a high temperature state, $S_1$ gives the extensive result for the thermal entropy that has nothing to do with entanglement). The Peres criterion [13, 14] is a very powerful starting point to quantify mixed state entanglement: it states that given a system described by the density matrix $\rho_A$, a sufficient condition for the presence of entanglement between two subsystems $A_1$ and $A_2$ (with $A = A_1 \cup A_2$) is that the partial transpose $\rho_A^{T_1}$ with respect to the degrees of freedom in $A_1$ (or equivalently $A_2$) has at least one negative eigenvalue. Starting from this criterion a *computable* measure of the bipartite entanglement for a general mixed state can be naturally defined as [15]

$$\mathcal{N} \equiv \frac{\mathrm{Tr}|\rho_A^{T_1}| - 1}{2}, \tag{2}$$

which is known as *negativity*. Here $\mathrm{Tr}|O| := \mathrm{Tr}\sqrt{O^\dagger O}$ denotes the trace norm of the operator $O$. Another equivalent measure, termed logarithmic negativity, has been also introduced in [15]

and it is defined as

$$\mathcal{E} \equiv \log \mathrm{Tr}|\rho_A^{T_1}|, \tag{3}$$

whose advantage with respect to $\mathcal{N}$ is that it scales and behaves more similarly to the Rényi entropies (indeed for pure states $\mathcal{E} = S_{1/2}$ [15]). Both $\mathcal{N}$ and $\mathcal{E}$ are entanglement monotones [15, 16]. It is also useful to define the moments of the partial transpose (a.k.a. the Rényi negativity, RN) as

$$R_n = \mathrm{Tr}(\rho_A^{T_1})^n. \tag{4}$$

Because of the non-positiveness of the spectrum of $\rho_A^{T_1}$, the Rényi negativities define two separate sequences for even and odd $n$. Then the natural way to exploit the replica trick is to obtain the negativity by considering the analytic continuation of the even sequence of $R_{n_e}$ at $n_e \to 1$ [17, 18] (which is different from $R_1 = 1$). The moments $R_n$ with integer $n \geq 2$ can also be measured in experiments [19–21], but they are not entanglement monotones. The entanglement negativity and Rényi negativities have been used to characterise mixed states in various quantum systems such as in harmonic oscillator chains [22–30], quantum spin models [31–44], (1+1)d conformal and integrable field theories [17, 18, 45–52], topologically ordered phases of matter in (2+1)d [53–57], out-of-equilibrium settings [20,58–66], holographic theories [67–72].

An interesting issue concerns the quantification of mixed state entanglement in fermionic systems. In particular, it has been pointed out that when $\rho_A$ is a Gaussian fermion operator, its partial transpose is the sum of two Gaussians; from this observation a procedure to extract the integer Rényi negativity was proposed [73] and was also used in many subsequent studies [74–80]. However, in this way the replica limit $n_e \to 1$ is not possible and hence, the negativity, i.e. the only genuine measure of entanglement, is not accessible. To overcome this problem, an alternative estimator of mixed state entanglement for fermionic systems has been introduced based on the time-reversal (TR) partial transpose (a.k.a *partial time reversal*) [81–88]. The new estimator has been dubbed *fermionic negativity*, although it is not related to negative eigenvalues of any matrix. It turned out that not only the fermionic negativity is an entanglement monotone [84], but also that it is able to detect entanglement in mixed states where the standard negativity vanishes. For both these reasons, throughout this work, we will mainly focus on the fermionic negativity.

In this manuscript, we consider a many-body system with an internal global symmetry and address the question of how mixed state entanglement splits into contributions arising from distinct symmetry sectors. The explicit idea of considering generally the internal structure of entanglement associated with symmetry is rather recent (the interested readers can consult the comprehensive literature on the subject [9,89–122]). For pure states, it has been established that the symmetry resolution of entanglement follows from the block diagonal form of the reduced density matrix [89,90]; one of the main findings is that the entanglement entropy is equally distributed among the different sectors [93]. For mixed states, the literature is limited to the pioneering work [91], where it was proven that whenever there is a conserved extensive charge, the negativity admits a resolution in terms of the charge imbalance between the two subsystems. Here, we first point out that by properly normalising the imbalance sectors (as also done in Ref. [121]) one obtains a clearer resolution of the entanglement in the imbalance; then we show that the imbalance-decomposition of negativity also holds using the partial TR definition for free fermions. We then use such decomposition to study the symmetry resolution of the entanglement of free fermions at finite temperature, exploiting the same field theory methods used for the total negativity [17,85].

The paper is organised as follows. In Section 2, we provide some basic definitions and briefly review the fermionic partial TR, motivating our work by simple examples for a tripartite and a bipartite geometry. After a brief summary of the results found in [91], we proceed with

the general definition of the imbalance operator and the consequent decomposition of the negativity, taking into account the normalisation of each sector. In Section 3, we review a method based on the replica trick to derive the leading order term for the Rényi entropy of massless Dirac fermions in (1+1)d when both the temperature $T$ and the system size $L$ are finite. As a warm-up, we use this method to compute the charged Rényi entropies. In Sections 4 and 5 we then provide results for charged and imbalance resolved negativity for tripartite and bipartite settings, respectively. Numerical checks for free fermions on the lattice are also presented as a benchmark of the analytical results. We draw our conclusions in Section 6. Four appendices are also included: they provide details about the analytical and numerical computations but they also make connections with some related ideas not developed here.

## 2  Charge imbalance resolved negativity

In this section, we briefly review the definition of partial time reversal for fermionic density matrices following Ref. [81]. Then we present the symmetry resolution of the standard partial transpose and of the partial TR of the density matrix. Simple examples will lead to a general definition of the imbalance resolution of entanglement negativity, both fermionic and bosonic. We closely follow Ref. [91], but we normalise differently the partial transpose in each symmetry sector, so that the symmetry resolved negativity is a genuine indicator of entanglement in the sector.

### 2.1  The fermionic partial time reversal density matrix

Let us start our discussion by recapitulating the definition of the partial transpose and its relation to the time-reversal transformation. Consider a density matrix $\rho_A$ in which $A$ is partitioned into two subsystems $A_1$ and $A_2$ such that $A = A_1 \cup A_2$ ($\rho_A$ can either be the reduced density matrix of a larger pure system $\rho_A = \text{Tr}_B(\rho)$ or a mixed density matrix, e.g. thermal, for an entire system). It can always be written as

$$\rho_A = \sum_{ijkl} \langle e_i^1, e_j^2 | \rho_A | e_k^1, e_l^2 \rangle \, |e_i^1, e_j^2\rangle \, \langle e_k^1, e_l^2|, \tag{5}$$

where $|e_j^1\rangle$ and $|e_k^2\rangle$ are orthonormal bases in the Hilbert spaces $\mathcal{H}_1$ and $\mathcal{H}_2$ corresponding to the $A_1$ and $A_2$ regions, respectively. The partial transpose of a density matrix for the subsystem $A_1$ is defined by exchanging the matrix elements in the subsystem $A_1$, i.e.

$$(|e_i^1, e_j^2\rangle \langle e_k^1, e_l^2|)^{T_1} \equiv |e_k^1, e_j^2\rangle \langle e_i^1, e_l^2|. \tag{6}$$

In terms of its eigenvalues $\lambda_i$, the trace norm of $\rho_A^{T_1}$ can be written as

$$\text{Tr}|\rho_A^{T_1}| = \sum_i |\lambda_i| = \sum_{\lambda_i > 0} |\lambda_i| + \sum_{\lambda_i < 0} |\lambda_i| = 1 + 2 \sum_{\lambda_i < 0} |\lambda_i|, \tag{7}$$

where in the last equality we used the normalisation $\sum_i \lambda_i = 1$. This expression makes evident that the negativity measures "how much" the eigenvalues of the partial transpose of the density matrix are negative, a property which is the reason for the name negativity. Moreover, in the absence of negative eigenvalues, $\text{Tr}|\rho_A^{T_1}| = 1$ and the negativity vanishes. For a bosonic system, it is known [14] that the partial transpose is the same as partial time reversal in phase space. This correspondence was exploited in harmonic chains to calculate the negativity in terms of the covariance matrix [22]. However, this is no longer true for fermions. To understand why, let us consider a single-site system described by fermionic operators $f$ and $f^\dagger$ which obey the

anticommutation relation $\{f, f^\dagger\} = 1$. We introduce the Grassmann variables $\xi, \bar{\xi}$, and the fermionic coherent states $|\xi\rangle = e^{-\xi f^\dagger} |0\rangle$ and $\langle \bar{\xi}| = \langle 0| e^{-f\bar{\xi}}$. In this basis, the time reversal transformation reads

$$|\xi\rangle \langle \bar{\xi}| \rightarrow |i\bar{\xi}\rangle \langle i\xi| \equiv (|\xi\rangle \langle \bar{\xi}|)^R. \tag{8}$$

This equation clearly shows that time reversal does not coincide with transposition because of the presence of the factor $i$. In the last equality we defined the time-reversal transpose, specified by the apex $R$ so as to distinguish it from the standard transposition for which we use the apex $T$. This transformation rule can be generalised to a many-particle (lattice) system with a partial TR only on the degrees of freedom within $A_1$ and reads

$$(|\{\xi_j\}_{j\in A_1}, \{\xi_j\}_{j\in A_2}\rangle \langle \{\bar{\chi}_j\}_{j\in A_1}, \{\bar{\chi}_j\}_{j\in A_2}|)^{R_1} = |\{i\bar{\chi}_j\}_{j\in A_1}, \{\xi_j\}_{j\in A_2}\rangle \langle \{i\xi_j\}_{j\in A_1}, \{\bar{\chi}_j\}_{j\in A_2}|, \tag{9}$$

where $|\{\xi_j\}\rangle = e^{-\sum_j \xi_j f_j^\dagger} |0\rangle$, $\langle \{\bar{\chi}_j\}| = \langle 0| e^{-\sum_j f_j \bar{\chi}_j}$ are the many-particle fermionic coherent states.

Let us consider the normal-ordered occupation number basis

$$|\{n_j\}_{j\in A_1}, \{n_j\}_{j\in A_2}\rangle = (f_{m_1}^\dagger)^{n_{m_1}} \dots (f_{m_{\ell_1}}^\dagger)^{n_{m_{\ell_1}}} (f_{m_1'}^\dagger)^{n_{m_1'}} \dots (f_{m_{\ell_2}'}^\dagger)^{n_{m_{\ell_2}'}} |0\rangle, \tag{10}$$

where $n_j$'s are occupation numbers in the subsystems $A_1$ and $A_2$, which have $\ell_1$ and $\ell_2$ sites respectively (in 1D they represent the lengths of intervals), and we use the indices $\{m_1, \dots, m_{\ell_1}\} \cup \{m_1', \dots, m_{\ell_2}'\}$ to denote the sites within the subsystem. The definition (9) in the occupation number basis is

$$(|\{n_j\}_{A_1}, \{n_j\}_{A_2}\rangle \langle \{\bar{n}_j\}_{A_1}, \{\bar{n}_j\}_{A_2}|)^{R_1} = (-1)^{\phi(\{n_j\}, \{\bar{n}_j\})} (|\{\bar{n}_j\}_{A_1}, \{n_j\}_{A_2}\rangle \langle \{n_j\}_{A_1}, \{\bar{n}_j\}_{A_2}|), \tag{11}$$

and can be viewed as the analogue of partial transposition in Eq. (6), up to the phase factor

$$\phi(\{n_j\}, \{\bar{n}_j\}) = \frac{[(\tau_1 + \bar{\tau}_1)\mathrm{mod}2]}{2} + (\tau_1 + \bar{\tau}_1)(\tau_2 + \bar{\tau}_2), \tag{12}$$

in which $\tau_s = \sum_{j\in A_s} n_j$, $\bar{\tau}_s = \sum_{j\in A_s} \bar{n}_j$ are the number of the occupied states in the $A_s$ intervals, $s = 1, 2$.

It is useful to rewrite the partial TR using the Majorana representation of the operator algebra. We introduce the Majorana operators as

$$c_{2j-1} = f_j + f_j^\dagger, \qquad c_{2j} = i(f_j - f_j^\dagger). \tag{13}$$

The density matrix in the Majorana representation takes the form

$$\rho_A = \sum_{\substack{\kappa, \tau \\ |\kappa|+|\tau|=\text{even}}} w_{\kappa, \tau} c_{m_1}^{\kappa_{m_1}} \dots c_{2m_{\ell_1}}^{\kappa_{2m_{\ell_1}}} c_{m_1'}^{\tau_{m_1'}} \dots c_{2m_{\ell_2}'}^{\tau_{2m_{\ell_2}'}}. \tag{14}$$

Here, $c_x^0 = \mathbb{I}$ and $c_x^1 = c_x$, $\kappa_i, \tau_j \in \{0, 1\}$ and $\kappa$ ($\tau$) is a $2m_{\ell_1}$-component vector ($2m_{\ell_2}'$) with norm $|\kappa| = \sum_j \kappa_j$ ($|\tau| = \sum_j \tau_j$). The constraint on the parity of $|\kappa| + |\tau|$ is due to the fact that the density matrix commutes with the total fermion-number parity operator, i.e. we focus our attention on physical states. Using Eq. (14), the partial TR with respect to the subsystem $A_1$ is defined by

$$\rho_A^{R_1} = \sum_{\substack{\kappa, \tau \\ |\kappa|+|\tau|=\text{even}}} i^{|\kappa|} w_{\kappa, \tau} c_{m_1}^{\kappa_{m_1}} \dots c_{2m_{\ell_1}}^{\kappa_{2m_{\ell_1}}} c_{m_1'}^{\tau_{m_1'}} \dots c_{2m_{\ell_2}'}^{\tau_{2m_{\ell_2}'}}. \tag{15}$$

We should note that the matrix resulting from the partial TR is not necessarily Hermitian and may have complex eigenvalues, although $\mathrm{Tr}\rho_A^{R_1} = 1$. Nevertheless, we can still use Eq.

(2) to define *a negativity* because the eigenvalues of the combined operator $[\rho_A^{R_1}(\rho_A^{R_1})^\dagger]$ are always real. Following Refs. [81, 85, 86], in the rest of the manuscript, we shall use the term negativity (or fermionic negativity) to refer to the quantity

$$\mathcal{N} \equiv \frac{\mathrm{Tr}|\rho_A^{R_1}| - 1}{2} = \frac{\mathrm{Tr}\sqrt{\rho_A^{R_1}(\rho_A^{R_1})^\dagger} - 1}{2}, \tag{16}$$

where the trace norm of the operator $\rho_A^{R_1}$ is the sum of the square roots of the eigenvalues of the product operator $\rho_A^{R_1}(\rho_A^{R_1})^\dagger$. Usually one also defines the (fermionic) Rényi negativities, as

$$R_n = \begin{cases} \mathrm{Tr}(\rho_A^{R_1}(\rho_A^{R_1})^\dagger \dots \rho_A^{R_1}(\rho_A^{R_1})^\dagger), & n \quad \text{even,} \\ \mathrm{Tr}(\rho_A^{R_1}(\rho_A^{R_1})^\dagger \dots \rho_A^{R_1}), & n \quad \text{odd,} \end{cases} \tag{17}$$

from which $\mathcal{N} = \frac{1}{2}\left(\lim_{n_e \to 1} R_{n_e} - 1\right)$, where $n_e$ denotes an even $n = 2m$ [81]. We stress that the fermionic negativity (16) is not related to the presence of negative eigenvalues in the spectrum of $\rho_A^{R_1}$. Sometimes, we will refer to the standard negativity (2) as the bosonic negativity.

## 2.2 Imbalance entanglement via bosonic partial transpose

In the presence of symmetries, the RDM has a block diagonal structure which allows to identify contributions to the entanglement entropy from individual charge sectors. In order to understand how symmetry is reflected in a block structure of the density matrix after partial transpose, we start with a simple example, taken from Ref. [91]. Consider a particle in one out of three boxes, $A_1, A_2, B$, described by a pure state $|\Psi\rangle = \alpha |100\rangle + \beta |010\rangle + \gamma |001\rangle$. The RDM of $A = A_1 \cup A_2$ is $\rho_A = \mathrm{Tr}_B |\psi\rangle\langle\psi| = |\gamma|^2 |00\rangle\langle 00| + (\alpha |10\rangle + \beta |01\rangle)(\alpha^* \langle 10| + \beta^* \langle 01|)$, i.e.

$$\rho_A = \begin{pmatrix} |\gamma|^2 & 0 & 0 & 0 \\ 0 & |\beta|^2 & \alpha^*\beta & 0 \\ 0 & \beta^*\alpha & |\alpha|^2 & 0 \\ 0 & 0 & 0 & 0 \end{pmatrix}, \tag{18}$$

in the basis $\{|00\rangle, |01\rangle, |10\rangle, |11\rangle\}$. This matrix is clearly block diagonal with respect to the total occupation number $N_A = N_1 + N_2$, where $N_1$ and $N_2$ respectively denote the particle number of the subsystem $A_1$ and $A_2$. According to Eq. (6), the partial transpose of $\rho_A$ is

$$\rho_A^{T_1} = \begin{pmatrix} |\gamma|^2 & 0 & 0 & \alpha\beta^* \\ 0 & |\beta|^2 & 0 & 0 \\ 0 & 0 & |\alpha|^2 & 0 \\ \beta\alpha^* & 0 & 0 & 0 \end{pmatrix}. \tag{19}$$

The total negativity is $\mathcal{N} = \left|\frac{1}{2}|\gamma|^2 - \sqrt{\frac{1}{4}|\gamma|^4 + |\alpha\beta|^2}\right|$. Once we reshuffle the elements of rows and columns in the basis of $\{|10\rangle, |00\rangle, |11\rangle, |01\rangle\}$, we get

$$\rho_A^{T_1} = \begin{pmatrix} |\alpha|^2 & 0 & 0 & 0 \\ 0 & |\gamma|^2 & \alpha\beta^* & 0 \\ 0 & \beta\alpha^* & 0 & 0 \\ 0 & 0 & 0 & |\beta|^2 \end{pmatrix}, \tag{20}$$

which has a block structure where each block is labelled by the occupation imbalance $q = N_2 - N_1$:

$$\rho_A^{T_1} \cong \left(|\alpha|^2\right)_{q=-1} \oplus \begin{pmatrix} |\gamma|^2 & \alpha\beta^* \\ \beta\alpha^* & 0 \end{pmatrix}_{q=0} \oplus \left(|\beta|^2\right)_{q=1}. \tag{21}$$

The structure of the above example is easily generalised to a many-body $\rho_A$ with subsystems $A_1$ and $A_2$ characterised by particle number operator $\hat{N}_1$ and $\hat{N}_2$; performing a partial transposition of the relation $[\rho_A, \hat{N}_A] = 0$ yields [91]

$$[\rho_A^{T_1}, \hat{N}_2 - \hat{N}_1^{T_1}] = 0, \tag{22}$$

from which we can do a block matrix decomposition according to the eigenvalues $q$ of the imbalance operator $\hat{Q} = \hat{N}_2 - \hat{N}_1^{T_1}$. We recall that this operator $\hat{Q}$ is basis dependent, as stressed in [91]; it has the form of an imbalance in the Fock basis (i.e. $\hat{Q} = \hat{N}_2 - \hat{N}_1$), while in others it can be different (as, e.g., in the computational basis we employ, $\hat{Q}$ is determined by the sum of the number operators up to some additive constants, see Appendix A and [91]).

Let $\mathcal{P}_q$ denote the projector onto the subspace of eigenvalue $q$ of the operator $\hat{Q}$. We define the *normalised charge imbalance partially transposed density matrix* as

$$\rho_A^{T_1}(q) = \frac{\mathcal{P}_q \rho_A^{T_1} \mathcal{P}_q}{\text{Tr}(\mathcal{P}_q \rho_A^{T_1})}, \qquad \text{Tr}(\rho_A^{T_1}(q)) = 1, \tag{23}$$

such that

$$\rho_A^{T_1} = \oplus_q p(q) \rho_A^{T_1}(q). \tag{24}$$

Here, $p(q) = \text{Tr}(\mathcal{P}_q \rho_A^{T_1})$ is the probability of finding $q$ as the outcome of a measurement of $\hat{Q}$ and corresponds to the sum of the diagonal elements of $\rho_A^{T_1}(q)$. Although the eigenvalues of $\rho_A^{T_1}$ can be negative, all the diagonal elements in the Fock basis are $\geq 0$ because the partial transpose leaves invariant all the elements on the diagonal and so they remain the same as those of $\rho_A$ which are $\geq 0$. This is evident in the example (20) and it is the same for any particle number. Hence $p(q)$ satisfies $p(q) \geq 0$ and $\sum_q p(q) = 1$, as it should be for a probability measure. We can thus define the (normalised) *charge imbalance resolved negativity* as

$$\mathcal{N}(q) = \frac{\text{Tr}|(\rho_A^{T_1}(q))| - 1}{2}. \tag{25}$$

Differently from [91], we prefer to deal with normalised quantities to preserve the natural meaning of negativity as a measure of entanglement: if in the $q$ sector there are no negative eigenvalues, according to Eq. (25), $\mathcal{N}(q) = 0$. Hence, this definition not only provides a resolution of the negativity, but also tells us in which sectors the negative eigenvalues are, i.e. where the entanglement is. The total negativity, $\mathcal{N}$, is resolved into (normalised) contributions from distinct imbalance sectors as

$$\mathcal{N} = \sum_q p(q) \mathcal{N}(q). \tag{26}$$

For the example of Eq. (21), the imbalance negativities are $\mathcal{N}(\pm 1) = 0$ and $\mathcal{N}(0) = \frac{1}{2}\left|1 - \sqrt{\frac{1}{2} + \left|\frac{2\alpha\beta}{|\gamma|^2}\right|^2}\right|$ with $p(0) = |\gamma|^2$; the only negative eigenvalue is in the sector $q = 0$. Eq. (26) gives back the total negativity. We stress that the imbalance decomposition of the negativity as in Eq. (26) cannot be performed for the logarithmic negativity in Eq. (3), because of the nonlinearity of the logarithm.

We conclude this section by discussing the important "pathological" case when $p(q) = 0$ for some values of the imbalance $q$, but $\rho_A^{T_1}(q)$ is non-zero and so the negativity of the sectors diverges, although the total one is finite. For example, this happens setting $\gamma = 0$ in Eq. (18); in this case $\rho_A$ corresponds to a pure state. Actually, it is obvious that every time that $\rho_A$ is a pure state there will be some $p(q) = 0$ because $N_1 + N_2$ is fixed and hence also the parity

of $N_1 - N_2$ is (so all the $p(q)$'s where $q$ has a different parity vanish). In such case, the origin of the problem can be traced back to the fact that the (pure-state) entanglement (entropy) is better resolved in terms of $N_1$ or $N_2$ rather than in the imbalance, i.e. the symmetries of $\rho_A$ and $\rho_A^{T_1}$ are larger than in the standard mixed case. However, mixed states with some zero $p(q)$ can be also easily built, although they are difficult to encounter as mixed states in physical settings (and they all correspond to states in which there is more symmetry than the imbalance). To understand the situation better, let us recall that $p(q)$ is always the sum of some diagonal elements of both $\rho_A^{T_1}(q)$ and $\rho_A$. For the latter, the diagonal elements are the populations of states in the Fock basis. Hence, we need at least a few zero populations to have a vanishing $p(q)$ (and, e.g., this will never happen in a Gibbs state at finite temperature). In the matrix $\rho_A$, if the populations in a given sector of the total charge are zero, the entire block is zero (and hence the entanglement entropy of the sector is zero). However, when taking the partial transpose, the off-diagonal elements are reshuffled in the matrix and, after being re-organised in terms of the imbalance, we can end up with some blocks with all zeros on the diagonal (and so $p(q) = 0$) but with non-zero off-diagonal elements. In these instances, we cannot normalise with $p(q)$. (Have always in mind the example of Eq. (18) with $\gamma = 0$: there are two sectors in $\rho_A$ with zero populations, $N_1 + N_2 = 0, 2$; after the partial transposition, they both end up in imbalance $q = 0$, see Eq. (21) which has non-zero off-diagonal terms). Anyhow, it makes sense that the imbalance negativity diverges in these cases. We are indeed facing sectors that have exactly zero populations, but still have some quantum correlations. In practice, as we shall see in the next section, these vanishing $p(q)$ are encountered only in the limit of a pure state (e.g. for $T \to 0$) and so diverging imbalance negativity signals that the state is getting pure and that a better resolution of the entanglement is in $N_1$ or $N_2$ rather than in the imbalance.

### 2.2.1 The example of tripartite CFT.

As a first simple example to show the importance of the normalisation $p(q)$ in the definition of imbalance resolved negativity, we reanalyse a simple known result [91] for the ground state of a Luttinger liquid (with parameter $K$) in a tripartite geometry. Thus, the results in this subsection describe gapless interacting 1d fermions. We focus on two adjacent intervals of length $\ell_1$ and $\ell_2$ respectively embedded in an infinite line.

Following [91], we start with the computation of the charged moments of the partial transpose

$$N_n^{T_1}(\alpha) \equiv \text{Tr}((\rho_A^{T_1})^n e^{i\hat{Q}\alpha}) = \langle \mathcal{T}_n \mathcal{V}_\alpha(u_1) \mathcal{T}_{-n}^2 \mathcal{V}_{-2\alpha}(v_1) \mathcal{T}_n \mathcal{V}_\alpha(v_2) \rangle, \tag{27}$$

where, in the rhs, we use the correspondence with the 3-point correlation function of fluxed twist field $\mathcal{T}_n \mathcal{V}_\alpha$ with scaling dimension

$$\Delta_n(\alpha) = \frac{1}{24}\left(n - \frac{1}{n}\right) + \frac{K}{2n}\left(\frac{\alpha}{2\pi}\right)^2, \qquad \Delta_{\mathcal{T}_{n_o}^2} = \Delta_{n_o} \qquad \Delta_{\mathcal{T}_{n_e}^2} = 2\Delta_{n_e/2}. \tag{28}$$

Using these scaling dimensions, one finds

$$\log N_n^{T_1}(\alpha) = \log R_n - \frac{K}{2n}\left(\frac{\alpha}{\pi}\right)^2 \log\left[\frac{\ell_1^2 \ell_2^2}{(\ell_1 + \ell_2)\epsilon^3}\right], \tag{29}$$

where $R_n$ are neutral Rényi negativities and $\epsilon$ is an ultraviolet cutoff. Notice in Eq. (29) only $R_n$ does depend on the parity of $n$ [17], while the $\alpha$ dependence is the same for even and odd $n$.

Upon performing a Fourier transform of Eq. (29), we obtain, through the saddle-point approximation, the (normalised) charge imbalance RN

$$R_n(q) = R_n \frac{\int_{-\pi}^{\pi} \frac{d\alpha}{2\pi} e^{-i(q-\bar{q})\alpha} e^{-\alpha^2 b_n/2}}{[\int_{-\pi}^{\pi} \frac{d\alpha}{2\pi} e^{-i(q-\bar{q})\alpha} e^{-\alpha^2 b_1/2}]^n} \simeq R_n \sqrt{\frac{(2\pi b_1)^n}{2\pi b_n}} e^{-\frac{(q-\bar{q})^2}{2}(\frac{1}{b_n} - \frac{n}{b_1})}, \tag{30}$$

where $\bar{q}$ is the expectation value of the charge operator $\hat{Q}$ and

$$b_n = \frac{1}{\pi^2 n} \log \left[ \frac{\ell_1^2 \ell_2^2}{(\ell_1 + \ell_2) \epsilon^3} \right]. \tag{31}$$

The saddle-point approximation holds for two intervals of length $\ell_1, \ell_2 \to \infty$ embedded in an infinite line at zero temperature. The replica limit $n_e \to 1$ is easily taken since there is no parity dependence in the imbalance part. For large $\ell_1, \ell_2 \to \infty$ (hence $b_n \to \infty$), we get

$$\mathcal{N}(q) = \mathcal{N} + o(1), \tag{32}$$

i.e. we found the equipartition of negativity in the different imbalance sectors at leading order for large subsystems. This behaviour is reminiscent of the equipartition of entanglement entropy in a pure quantum system that possesses an internal symmetry [93]. It is clear that *negativity equipartition* can be shown only by properly normalising the partial transpose in each sector as done here. As an important difference compared to the entanglement entropies, we do not have additional $\log \log \ell$ [96] corrections to the symmetry resolved quantities.

## 2.3 Imbalance entanglement of fermions via partial TR

Now we are ready to understand the block structure of the partial TR density matrix and how the fermionic negativity splits according to the symmetry. We first revisit the simple example of the previous section in Eq. (18) for fermions. According to Eq. (8), the partial TR of $\rho_A$ in Eq. (18) is

$$\rho_A^{R_1} = \begin{pmatrix} |\gamma|^2 & 0 & 0 & i\alpha\beta^* \\ 0 & |\beta|^2 & 0 & 0 \\ 0 & 0 & |\alpha|^2 & 0 \\ i\beta\alpha^* & 0 & 0 & 0 \end{pmatrix}, \tag{33}$$

i.e. the partial TR transformation does not spoil the block matrix structure according to the occupation imbalance $q = N_2 - N_1$:

$$\rho_A^{R_1} \cong \left( |\alpha|^2 \right)_{q=-1} \oplus \begin{pmatrix} |\gamma|^2 & i\alpha\beta^* \\ i\beta\alpha^* & 0 \end{pmatrix}_{q=0} \oplus \left( |\beta|^2 \right)_{q=1}. \tag{34}$$

The (total) fermionic negativity is

$$\mathcal{N} = \frac{|\gamma|^2}{2} \left( -1 + \sqrt{\frac{1}{2} + \frac{|\alpha\beta|^2}{|\gamma|^4}} + \sqrt{\frac{1}{4} + \frac{|\alpha\beta|^2}{|\gamma|^4}} + \sqrt{\frac{1}{2} + \frac{|\alpha\beta|^2}{|\gamma|^4} - \sqrt{\frac{1}{4} + \frac{|\alpha\beta|^2}{|\gamma|^4}}} \right). \tag{35}$$

For a many-body state, the analogue of the commutation relation in Eq. (22) now reads

$$[\rho_A^{R_1}, \hat{N}_2 - \hat{N}_1^{R_1}] = 0, \tag{36}$$

while the (normalised) charge imbalance resolved negativity is given by

$$\mathcal{N}(q) = \frac{\text{Tr}|(\rho_A^{R_1}(q))| - 1}{2}, \qquad \rho_A^{R_1}(q) = \frac{\mathcal{P}_q \rho_A^{R_1} \mathcal{P}_q}{\text{Tr}(\mathcal{P}_q \rho_A^{R_1})}. \tag{37}$$

We also define the *charge imbalance resolved RN*

$$R_n(q) = \begin{cases} \text{Tr}(\rho_A^{R_1}(q)\rho_A^{R_1}(q)^\dagger \dots \rho_A^{R_1}(q)\rho_A^{R_1}(q)^\dagger), & n \quad \text{even}, \\ \text{Tr}(\rho_A^{R_1}(q)\rho_A^{R_1}(q)^\dagger \dots \rho_A^{R_1}(q)), & n \quad \text{odd}, \end{cases} \tag{38}$$

from which $\mathcal{N}(q) = \frac{1}{2}\left(\lim_{n_e \to 1} R_{n_e}(q) - 1\right)$. It is important to stress that the diagonal elements of $\rho_A^{R_1}$ are the same as $\rho_A^{T_1}$ (the TR operation does not touch the diagonal elements) and so the probabilities $p(q)$ are identical for both the standard and the TR partial transpose. Thus, all the considerations for the vanishing of $p(q)$ in the previous subsection apply also here. For the example of Eq. (34), the imbalance negativities are $\mathcal{N}(\pm 1) = 0$ and $\mathcal{N}(0) = \frac{1}{2}\left(-1 + \sqrt{\frac{1}{2} + \frac{|\alpha\beta|^2}{|\gamma|^4}} + \sqrt{\frac{1}{4} + \frac{|\alpha\beta|^2}{|\gamma|^4}} + \sqrt{\frac{1}{2} + \frac{|\alpha\beta|^2}{|\gamma|^4} - \sqrt{\frac{1}{4} + \frac{|\alpha\beta|^2}{|\gamma|^4}}}\right)$ with $p(0) = |\gamma|^2$. As a further check, $\sum_q p(q)\mathcal{N}(q)$ gives back the total negativity in Eq. (35).

We think it is beneficial to give another basic example (taken from Ref. [85]) of imbalance resolution with free fermions on a two-site lattice model described by the Hamiltonian

$$\hat{H} = -\Delta(f_1^\dagger f_2 + f_2^\dagger f_1), \tag{39}$$

where $\Delta$ is a tunnelling amplitude. In the basis $\{|00\rangle, |01\rangle, |10\rangle, |11\rangle\}$, the thermal density matrix is

$$\rho = \frac{e^{-\beta\hat{H}}}{\text{Tr}(e^{-\beta\hat{H}})} = \frac{1}{2 + 2\cosh(\beta\Delta)}\begin{pmatrix} 1 & 0 & 0 & 0 \\ 0 & \cosh(\beta\Delta) & \sinh(\beta\Delta) & 0 \\ 0 & \sinh(\beta\Delta) & \cosh(\beta\Delta) & 0 \\ 0 & 0 & 0 & 1 \end{pmatrix}. \tag{40}$$

Let us take the partial TR

$$\rho^{R_1} = \frac{1}{2 + 2\cosh(\beta\Delta)}\begin{pmatrix} 1 & 0 & 0 & i\sinh(\beta\Delta) \\ 0 & \cosh(\beta\Delta) & 0 & 0 \\ 0 & 0 & \cosh(\beta\Delta) & 0 \\ i\sinh(\beta\Delta) & 0 & 0 & 1 \end{pmatrix}. \tag{41}$$

By reshuffling the elements of rows and columns in the basis of $\{|00\rangle, |11\rangle, |10\rangle, |01\rangle\}$, $\rho^{R_1}$ has a block matrix structure in the occupation imbalance between the subsystem and the rest of the system that we can write explicitly as

$$\rho^{R_1} \cong \left(\frac{\cosh(\beta\Delta)}{2 + 2\cosh(\beta\Delta)}\right)_{q=-1} \oplus \begin{pmatrix} \frac{1}{2+2\cosh(\beta\Delta)} & \frac{i\sinh(\beta\Delta)}{2+2\cosh(\beta\Delta)} \\ \frac{i\sinh(\beta\Delta)}{2+2\cosh(\beta\Delta)} & \frac{1}{2+2\cosh(\beta\Delta)} \end{pmatrix}_{q=0} \oplus \left(\frac{\cosh(\beta\Delta)}{2+2\cosh(\beta\Delta)}\right)_{q=1}. \tag{42}$$

When the state becomes pure, i.e. $\beta\Delta \gg 1$, $p(q=0) \to 0$. The interpretation is the same as the one for the bosonic negativity: when the state is pure, the operator to resolve the symmetry is $\hat{N}_1$ (or $\hat{N}_2$), rather than the imbalance. For completeness, we report the fermionic negativity

$$\mathcal{N} = \frac{1}{2}\tanh^2\left(\frac{\beta\Delta}{2}\right), \tag{43}$$

and its splitting in the imbalance sectors: $\mathcal{N}(\pm 1) = 0$ and $\mathcal{N}(0) = \frac{1}{2}(\cosh(\beta\Delta) - 1)$ with $p(0) = \frac{1}{\cosh(\beta\Delta)+1}$, so that $\sum_q p(q)\mathcal{N}(q) = p(0)\mathcal{N}(0) = \mathcal{N}$. Notice that as $\beta \to \infty$, $p(0) \to 0$, $\mathcal{N}(0) \to \infty$, but their product stays finite and tends to 1/2.

# 3 Replica approach

In this section, we first review the replica approach to the charged entropies [118] and apply it to their calculation for a massless Dirac fermion at finite temperature, a result that was not yet obtained so far. Then we adapt the method to the charged Rényi negativities. Its applications will be presented in the successive sections.

## 3.1 Charged moments of the reduced density matrix

We start by recalling the symmetry resolution of the entanglement entropy. As already mentioned, in the presence of a $U(1)$ symmetry $\hat{Q}$, $\rho_A$ admits a charge decomposition according to the local charge $\hat{Q}_A$, where each block corresponds to different eigenspaces of $\hat{Q}_A$, which we can label as $\tilde{q} \in \mathbb{Z}$, i.e.

$$\rho_A = \oplus_{\tilde{q}} \tilde{p}(\tilde{q}) \rho_A(\tilde{q}), \quad \tilde{p}(\tilde{q}) = \mathrm{Tr}(\mathcal{P}_{\tilde{q}} \rho_A). \tag{44}$$

Here we use $\tilde{q}$ for the eigenvalues of $\hat{Q}_A$ to make a clear distinction with the eigenvalues of the imbalance $q$. Unless differently specified, $A$ is a generic subsystem made of $p$ intervals $[u_i, v_i]$, i.e $A = \cup_{i=1}^{p}[u_i, v_i]$. The *symmetry resolved Rényi entropies* are then defined as [93]

$$S_n(\tilde{q}) \equiv \frac{1}{1-n} \log \mathrm{Tr}[\rho_A(\tilde{q})]^n. \tag{45}$$

The direct use of the above definition to evaluate the symmetry resolved entropy requires the knowledge of the spectrum of the RDM and its resolution in $\tilde{q}$, that is a nontrivial problem, especially for analytic computations. However, we can use the Fourier representation of the projection operator and focus on the *charged moments* of $\rho_A$, $Z_n(\alpha) \equiv \mathrm{Tr}[\rho_A^n e^{i\alpha \hat{Q}_A}]$ [90]. Their Fourier transforms

$$\mathcal{Z}_n(\tilde{q}) = \int_{-\pi}^{\pi} \frac{d\alpha}{2\pi} e^{-i\tilde{q}\alpha} Z_n(\alpha), \tag{46}$$

are related to the entropies of the sector of charge $\tilde{q}$ as

$$S_n(\tilde{q}) = \frac{1}{1-n} \log\left[ \frac{\mathcal{Z}_n(\tilde{q})}{\mathcal{Z}_1(\tilde{q})^n} \right]. \tag{47}$$

We exploit the framework of the replica trick to evaluate the charged moments, which are the main object of interest in this section.

In a generic quantum field theory, the replica trick for computing $Z_n(\alpha)$ can be implemented by inserting an Aharonov-Bohm flux through a multi-sheeted Riemann surface $\mathcal{R}_n$, such that the total phase accumulated by the field upon going through the entire surface is $\alpha$ [90]. The result is that $Z_n(\alpha)$ is the partition function of such a modified surface, that, following Ref. [90], we dub $\mathcal{R}_{n,\alpha}$. Here we focus on a massless Dirac fermion described by the Lagrangian density

$$\mathcal{L} = \bar{\Psi} \gamma^\mu \partial_\mu \Psi, \tag{48}$$

where $\bar{\Psi} = \Psi^\dagger \gamma^0$, $\gamma^0 = \sigma^1$, $\gamma^1 = \sigma^2$. Rather than dealing with fields defined on a non trivial manifold $\mathcal{R}_{n,\alpha}$, it is more convenient to work on a single plane with a $n$-component field

$$\Psi = \begin{pmatrix} \psi_1 \\ \psi_2 \\ \vdots \\ \psi_n \end{pmatrix}, \tag{49}$$

where $\psi_j$ is the field on the $j$-th copy. Upon crossing the cut $A$, the vector field $\Psi$ transforms according to the twist matrix $T_\alpha$

$$
T_\alpha = \begin{pmatrix} 0 & e^{i\alpha/n} & & \\ & 0 & e^{i\alpha/n} & \\ & & \ddots & \ddots \\ (-1)^{n-1}e^{i\alpha/n} & & & 0 \end{pmatrix}.
\tag{50}
$$

The idea of using the twist matrix for the Dirac fermions at $\alpha = 0$ was originally suggested in [123] (see also [124]). The matrix $T_\alpha$ has eigenvalues

$$
\lambda_k = e^{i\frac{\alpha}{n}}e^{2\pi i\frac{k}{n}}, \quad k = -\frac{n-1}{2}, \ldots, \frac{n-1}{2}.
\tag{51}
$$

By diagonalising $T_\alpha$ with a unitary transformation, the problem is reduced to $n$ decoupled and multi-valued fields $\psi_k$ in a two dimensional spacetime. This technique is applicable only to free theories, otherwise the $k-$modes do not decouple. In particular, the charged moments become

$$
Z_n(\alpha) = \prod_{k=-(n-1)/2}^{(n-1)/2} Z_{k,n}(\alpha),
\tag{52}
$$

where $Z_{k,n}(\alpha)$ is the partition function for a Dirac field that along $A$ picks up a phase equal to $e^{i\frac{\alpha}{n}}e^{2\pi i\frac{k}{n}}$, or equivalently the phase picked up going around one of the entangling points $u_i, v_i$ is $e^{i\frac{\alpha}{n}}e^{2\pi i\frac{k}{n}}$ and $e^{-i\frac{\alpha}{n}}e^{-2\pi i\frac{k}{n}}$, respectively. The main difference with respect to the standard computation for the Rényi entropies is that, for a charged quantity, the boundary conditions of the multivalued fields along $A$ depend also on the flux $\alpha$ and not only on the replica index. This multivaluedness can be circumvented with the same trick used for $\alpha = 0$ [123], i.e. by absorbing it in an *external* gauge field coupled to a single-valued fields $\tilde{\psi}_k$. Indeed, the singular gauge transformation

$$
\psi_k(x) = e^{i\oint_C dy_\mu A_k^\mu(y)}\tilde{\psi}_k(x),
\tag{53}
$$

allows us to absorb the phase along $A$ into the gauge field at the price of changing the Lagrangian density into

$$
\mathcal{L}_k = \bar{\tilde{\psi}}_k \gamma^\mu(\partial_\mu + iA_\mu^k)\tilde{\psi}_k.
\tag{54}
$$

The actual value of $A_\mu^k$ in Eq. (53) is fixed by requiring that, for any loop $C$, the original boundary conditions for the multivalued field $\psi_k$ are reproduced. This is achieved with

$$
\begin{aligned}
\oint_{C_{u_i}} dx^\mu A_\mu^k &= -\frac{2\pi k}{n} - \frac{\alpha}{n}, \\
\oint_{C_{v_i}} dx^\mu A_\mu^k &= +\frac{2\pi k}{n} + \frac{\alpha}{n},
\end{aligned}
\tag{55}
$$

where $C_{u_i}$ and $C_{v_i}$ are circuits around left and right endpoints of the $i$-th interval. If the circuit $C$ does not encircle any endpoint, $\oint_C dx^\mu A_\mu^k = 0$. If more endpoints are encircled the phases sum up. It is useful to rewrite Eq. (55) in the corresponding differential form, i.e. using Stokes' theorem

$$
\epsilon^{\mu\nu}\partial_\nu A_\mu^k(x) = 2\pi\left(\frac{k}{n} + \frac{\alpha}{2\pi n}\right)\sum_{i=1}^{p}[\delta(x-u_i) - \delta(x-v_i)],
\tag{56}
$$

where $p$ is the number of intervals.

After the transformation (53), the desired charged partition sum $Z_{k,n}(\alpha)$ is written as

$$Z_{k,n}(\alpha) = \langle e^{i \int d^2 x A^k_\mu j^\mu_k} \rangle, \tag{57}$$

where $j^\mu_k = \bar{\tilde{\psi}}_k \gamma^\mu \tilde{\psi}_k$ is the Dirac current and $A^k_\mu$ satisfies Eq. (55) or, equivalently, (56). Eq. (57) is more easily calculated by bosonisation, which maps the Dirac current to the derivative of a scalar field and the Lagrangian of the $k$-th fermion to that of a real massless scalar field $\phi_k$, $\mathcal{L}_k = \frac{1}{8\pi} \partial_\mu \phi_k \partial^\mu \phi_k$ (here we work with the normalisation of the boson field such that the Dirac fermion corresponds to a compactified boson with radius $R = 2$, as in [125]). Therefore we can evaluate $Z_{k,n}(\alpha)$ as the correlation function of the vertex operators $V_a(x) = e^{-ia\phi_k(x)}$, i.e.

$$Z_{k,n}(\alpha) = \langle \prod_{i=1}^{p} V_{\frac{k}{n}+\frac{\alpha}{2\pi n}}(u_i) V_{-\frac{k}{n}-\frac{\alpha}{2\pi n}}(v_i) \rangle. \tag{58}$$

An important observation about Eq. (55) is that we can arbitrarily add $2\pi m$ phase shifts, with $m$ an integer, to the right hand side without affecting the total phase factor along the circuits $\mathcal{C}_{u_i}$ and $\mathcal{C}_{v_i}$ defined above. This ambiguity leads to inequivalent different representations of the partition function $Z_{k,n}(\alpha)$ in Eq. (52), which in turn must be written as a summation over all allowed representations. The asymptotic behaviour of each term for large subsystem size, $\ell$, is a power law $\ell^{-\alpha_m}$ and the leading term corresponds to the one with the smallest exponent $\alpha_m$. For the charged moments, the leading order is given by $m = 0$, but this is not the case for the entanglement negativity. See Appendix B for a more detailed discussion of this issue.

Let us now apply this machinery to study the charged moments of a free Dirac fermion on a torus with multiple intervals $(u_a, v_a)$, $(a = 1, \ldots, p)$. To have more compact formulas, we rescale the spatial coordinates by the system size $L$. The torus is defined by two periods which, in our units, are 1 and $\tau = i\beta/L$, where $\beta = 1/T$ is the inverse temperature. The partition function depends on the boundary conditions along the two cycles, which specify the spin structure of the fermion on the torus. Let $z$ be a holomorphic coordinate on the torus: it has the periodicities $z = z + 1$ and $z = z + \tau$. The holomorphic component of the fermion on the torus satisfies four possible boundary conditions

$$\tilde{\psi}_k(z+1) = e^{2\pi i \nu_1} \tilde{\psi}_k(z), \qquad \tilde{\psi}_k(z+\tau) = e^{2\pi i \nu_2} \tilde{\psi}_k(z), \tag{59}$$

where $\nu_1$ and $\nu_2$ take the values 0 or $\frac{1}{2}$. The anti-holomorphic component is a function of $\bar{z}$ and satisfies the same boundary conditions as the holomorphic part. We denote the $\nu = (\nu_1, \nu_2)$ sector where $\nu = 1, 2, 3, 4$ corresponds to $(0,0), (0,1/2), (1/2,1/2), (1/2,0)$, respectively (for standard fermions, the physical boundary conditions are anti-periodic along both cycles and so $\nu = 3$, but the other spin structures have important applications too). Hence, we just need the correlation function of the vertex operators $V_e(z, \bar{z}) = e^{ie\phi(z,\bar{z})}$ on the torus with boundary conditions corresponding to the sector $\nu$. These can be found in Ref. [125] and read

$$\langle V_{e_1}(z_1, \bar{z}_1) V_{e_2}(z_2, \bar{z}_2) \ldots V_{e_N}(z_N, \bar{z}_N) \rangle_\nu = \left| \prod_{i<j} \frac{\partial_z \theta_1(0|\tau)}{\theta_1(z_i - z_j|\tau)} \right|^{-2e_i e_j} \left| \frac{\theta_\nu(\sum_i (e_i z_i)|\tau)}{\theta_\nu(0|\tau)} \right|^2. \tag{60}$$

In Eq. (60) and afterwards, we use the notation $\partial_z \theta_1(0|\tau) = \partial_z \theta_1(z|\tau)|_{z=0}$. Plugging Eq. (60) into Eq. (58), we have in sector $\nu$

$$Z^{(\nu)}_{k,n}(\alpha) = \left| \frac{\prod_{i<j} \theta_1(u_i - u_j|\tau) \theta_1(v_i - v_j|\tau)}{\prod_{i,j} \theta_1(u_i - v_j|\tau)} \left( \frac{\epsilon}{L} \partial_z \theta_1(0|\tau) \right)^p \right|^{2(\frac{k}{n}+\frac{\alpha}{2\pi n})^2} \times$$

$$\left| \frac{\theta_\nu((\frac{k}{n}+\frac{\alpha}{2\pi n}) \sum_i (u_i - v_i)|\tau)}{\theta_\nu(0|\tau)} \right|^2, \tag{61}$$

where $\epsilon$ is an ultraviolet cutoff which depends on both $\alpha$ and $n$, although we almost always omit such a dependence for conciseness. The total charged moments are finally obtained by taking the product (52) to get

$$\log Z_n^{(\nu)}(\alpha) = \log Z_{n,0}(\alpha) + \log Z_{n,1}^{(\nu)}(\alpha), \tag{62}$$

where the first term is spin-independent

$$\log Z_{n,0}(\alpha) = \left[\frac{1}{6}\left(n - \frac{1}{n}\right) + \frac{\alpha^2}{2\pi^2 n}\right] \log\left|\frac{\prod_{i<j}\theta_1(u_i - u_j|\tau)\theta_1(v_i - v_j|\tau)}{\prod_{i,j}\theta_1(u_i - v_j|\tau)}\left(\frac{\epsilon}{L}\partial_z\theta_1(0|\tau)\right)^p\right|$$

$$\equiv \log Z_{n,0}(0) - \frac{\alpha^2}{2}\mathcal{B}_n^0, \tag{63}$$

(in the second line we implicitly defined $\mathcal{B}_n^0$) while the second one depends on the sector $\nu$

$$\log Z_{n,1}^{(\nu)}(\alpha) = 2 \sum_{k=-(n-1)/2}^{(n-1)/2} \log\left|\frac{\theta_\nu((\frac{k}{n} + \frac{\alpha}{2\pi n})\sum_i(u_i - v_i)|\tau)}{\theta_\nu(0|\tau)}\right|. \tag{64}$$

The Fourier transform of the charged moments (62) gives the symmetry resolved moments and entropies. We report only the results for $\nu = 3$, but similarly also the others may be obtained. Using the product representation of the theta functions

$$\theta_3(z|\tau) = \prod_{m=1}^{\infty}(1 - e^{2\pi i\tau m})(1 + e^{2\pi iz}e^{2\pi i\tau(m-1/2)})(1 + e^{-2\pi iz}e^{2\pi i\tau(m-1/2)}), \tag{65}$$

the sum over $k$ of the spin-dependent term in Eq. (64) can be explicitly worked out as

$$\log Z_{n,1}^{(3)}(\alpha) = 2\sum_{j\geq 1}\frac{(-1)^j}{j\sinh(\pi j\beta/L)}\left(n - \cos\left(\frac{\alpha rj}{n}\right)\frac{\sin(\pi jr)}{\sin(\frac{\pi jr}{n})}\right), \tag{66}$$

where $r = \sum_i(u_i - v_i)$. Since the symmetry resolved entropies will be obtained from a saddle point, we expand at the second order in $\alpha$, obtaining

$$\log Z_{n,1}^{(3)}(\alpha) \simeq$$
$$2\sum_{j\geq 1}\frac{(-1)^j}{j\sinh(\pi j\beta/L)}\left(n - \frac{\sin(\pi jr)}{\sin(\frac{\pi jr}{n})}\right) + \frac{\alpha^2 r^2}{n^2}\sum_{j\geq 1}\frac{(-1)^j j}{\sinh(\pi j\beta/L)}\frac{\sin(\pi jr)}{\sin(\frac{\pi jr}{n})} = \mathcal{A}_n - \frac{\alpha^2}{2}\mathcal{B}_n^1. \tag{67}$$

The sum converges very fast in $j$ and very few terms are sufficient to get it. Introducing $\mathcal{B}_n \equiv \mathcal{B}_n^0 + \mathcal{B}_n^1$, the charged moments of the RDM are

$$Z_n^{(3)}(\alpha) = Z_n^{(3)}(0)e^{-\frac{\alpha^2}{2}\mathcal{B}_n}, \tag{68}$$

with Fourier transform

$$\mathcal{Z}_n^{(3)}(\tilde{q}) = Z_n^{(3)}(0)\int_{-\pi}^{\pi}\frac{d\alpha}{2\pi}e^{-i\alpha\tilde{q}}e^{-\frac{\alpha^2}{2}\mathcal{B}_n} \simeq \frac{Z_n^{(3)}(0)}{\sqrt{2\pi\mathcal{B}_n}}e^{-\frac{\tilde{q}^2}{2\mathcal{B}_n}}, \tag{69}$$

where we exploited the saddle point approximation and we used that $\bar{\tilde{q}}$, the expectation value of the charge operator $\hat{Q}_A$, vanishes for a free Dirac field at any temperature. From the definition (47), we get the symmetry resolved Rényi entropies

$$S_n(\tilde{q}) = S_n - \frac{1}{2}\log(2\pi\mathcal{B}_1) + O(1). \tag{70}$$

In order to make contact with some known results in literature [93, 96, 98], we report the explicit expression for $\mathcal{B}_1$ in the low and high temperature limit for $p = 1$

$$
\mathcal{B}_1 = \begin{cases} \frac{1}{\pi^2} \log\left(\frac{L}{\pi\epsilon} \sin\frac{\ell\pi}{L}\right), & LT \ll 1, \\ \frac{1}{\pi^2} \log\left(\frac{\beta}{\pi\epsilon} \sinh\frac{\ell\pi}{\beta}\right), & LT \gg 1. \end{cases} \tag{71}
$$

The result in Eq. (70) has been dubbed equipartition of entanglement [93]: at leading order the entanglement is the same in the different charge sectors. In [93] this was proven for gapless interacting 1d fermions at zero temperature. As a side result, here we showed that entanglement equipartition holds also at finite size and temperature, at least for a free Dirac field.

## 3.2  Charged moments of the partial transpose

The above procedure is easily adapted to the computation of the charged moments of the partial TR defined as

$$
N_n(\alpha) = \begin{cases} \text{Tr}(\rho_A^{R_1}(\rho_A^{R_1})^\dagger \dots \rho_A^{R_1}(\rho_A^{R_1})^\dagger e^{i\hat{Q}_A\alpha}), & n \quad \text{even}, \\ \text{Tr}(\rho_A^{R_1}(\rho_A^{R_1})^\dagger \dots \rho_A^{R_1} e^{i\hat{Q}_A\alpha}), & n \quad \text{odd}. \end{cases} \tag{72}
$$

Hence, in order to compute the imbalance resolved negativity, we need to study the composite operator $\rho_A^{R_1}(\rho_A^{R_1})^\dagger$. The charged moments in Eq. (72) are defined for two subsystems $A_1$ and $A_2$ with different twist matrices respectively denoted by $T_\alpha^{R_1}$ and $T_\alpha$. The new twist matrix $T_\alpha^{R_1}$ for the transposed time reversed subsystem is given by

$$
T_\alpha^{R_1} = \begin{pmatrix} 0 & 0 & \dots & (-1)^{n-1}e^{-i\alpha/n} \\ e^{-i\alpha/n} & 0 & & \\ 0 & e^{-i\alpha/n} & & \ddots \\ & & \ddots & \ddots \end{pmatrix}. \tag{73}
$$

The two matrices, $T_\alpha$ and $T_\alpha^{R_1}$, are simultaneously diagonalisable. Consequently, we can decompose our problem into $n$ decoupled copies in which the fields have different twist phases along the two subsystems. As a result, $N_n(\alpha)$ is decomposed as

$$
N_n(\alpha) = \prod_{k=-(n-1)/2}^{(n-1)/2} Z_{R_1,k}(\alpha), \tag{74}
$$

where $Z_{R_1,k}(\alpha)$ is the partition function for fields with twist phases equal to $e^{-2\pi i\left(\frac{k}{n} + \frac{\alpha}{2\pi n}\right)}$ and $e^{2\pi i\left(\frac{k}{n} + \frac{\alpha}{2\pi n} - \frac{\varphi_n}{2\pi}\right)}$, respectively along $A_2$ and $A_1$. Here $\varphi_n = \pi$ for $n = n_e$ even and $\varphi_n = \frac{n-1}{n}\pi$ for $n = n_o$ odd (as follows from the diagonalisation of Eq. (73)). In particular, the probability $p(q)$ is the Fourier transform of $N_1(\alpha) = \text{Tr}[\rho^{R_1}e^{i\hat{Q}\alpha}]$, that, with a minor abuse of terminology, we dub *charged probability*. In this case, the twist matrices along the two intervals are just phases given by $T_\alpha = e^{i\alpha}$ and $T_\alpha^{R_1} = T_\alpha^{-1} = e^{-i\alpha}$. For a system of interacting fermions (i.e. for a free compact boson with different compactification radius), the procedure outlined here does not apply. The calculation is much more cumbersome and requires to adapt the technique for the standard negativity (see [18]) to the PT case, but this has not yet been done even for the total negativity.

A Fourier transform leads us to the imbalance resolved negativities (38)

$$
\mathcal{Z}_{R_1,n}(q) = \int_{-\pi}^{\pi} \frac{d\alpha}{2\pi} e^{-i\alpha q} N_n(\alpha), \quad p(q) = \int_{-\pi}^{\pi} \frac{d\alpha}{2\pi} e^{-i\alpha q} N_1(\alpha), \tag{75}
$$



Figure 1: Tripartite geometry for two adjacent intervals. In the plot the interval $A_1$ is the blue one on the left and $A_2$ the grey one on the right, of length $\ell_1$ and $\ell_2$ respectively. $B$ is the reminder. The partial transpose is taken on $A_1$. For each branch point, we report the phase taken by the field $\psi_k$ going around it.

from which

$$R_n(q) = \frac{\mathcal{Z}_{R_1,n}(q)}{p^n(q)}, \qquad \mathcal{N}(q) = \frac{1}{2}\Big( \lim_{n_e \to 1} R_{n_e}(q) - 1 \Big). \tag{76}$$

Let us stress the replica limit for $R_{n_e}(q)$ is $\lim_{n_e \to 1} \frac{\mathcal{Z}_{R_1,n_e}(q)}{p(q)}$, i.e. while it is sufficient to set $n = 1$ in the denominator, the numerator requires an analytic continuation from the even sequence at $n_e \to 1$, in agreement with the definition in Eq. (16). In the following section, we compute the imbalance resolved entanglement negativity for two geometries.

## 4 Charged and symmetry resolved negativities in a tripartite geometry

Let us study the negativity of two subsystems consisting of two adjacent intervals $A_1$, $A_2$, of lengths $\ell_1$, $\ell_2$ out of a system of length $L$, as depicted in Fig 1. We place the branch points at $u_1 = -\ell_1/L = -r_1, v_1 = u_2 = 0$, and $v_2 = \ell_2/L = r_2$ and the multivalued fields $\psi_k$ take up a phase $e^{2\pi i(\frac{k}{n} + \frac{\alpha}{2\pi n} - \frac{\varphi_n}{2\pi})}$ at $u_1$, $e^{-2\pi i(\frac{k}{n} + \frac{\alpha}{2\pi n} - \frac{\varphi_n}{2\pi})} e^{-2\pi i(\frac{k}{n} + \frac{\alpha}{2\pi n})}$ at $v_1$ and $e^{2\pi i(\frac{k}{n} + \frac{\alpha}{2\pi n})}$ at $v_2$. By introducing a gauge field $A_\mu^k$, as explained in Sec. 3.1, we have to impose proper monodromy conditions such that the field is almost pure gauge except at the branch points, where delta function singularities are necessary to recover the correct phases of the multivalued fields. Hence, the flux of the gauge fields is given by

$$\frac{1}{2\pi} \epsilon^{\mu\nu} \partial_\nu A_\mu^k(x) = \Big(\frac{k}{n} + \frac{\alpha}{2\pi n} - \frac{\varphi_n}{2\pi}\Big)\delta(x - u_1) - \Big(\frac{2k}{n} + \frac{\alpha}{\pi n} - \frac{\varphi_n}{2\pi}\Big)\delta(x - v_1) + \Big(\frac{k}{n} + \frac{\alpha}{2\pi n}\Big)\delta(x - v_2). \tag{77}$$

Through bosonisation, $Z_{R_1,k}^{(\nu)}(\alpha)$ can be written as a correlation function of vertex operators $V_a(x) = e^{-ia\phi_k(x)}$ as

$$\begin{aligned} Z_{R_1,k}^{(\nu)}(\alpha) &= \Big\langle V_{\frac{k}{n} + \frac{\alpha}{2\pi n} - \frac{\varphi_n}{2\pi}}(u_1) V_{-\frac{k}{n} - \frac{\alpha}{2\pi n} + \frac{\varphi_n}{2\pi}}(v_1) V_{-\frac{k}{n} - \frac{\alpha}{2\pi n}}(v_1) V_{\frac{k}{n} + \frac{\alpha}{2\pi n}}(v_2) \Big\rangle \\ &= \Big\langle V_{\frac{k}{n} + \frac{\alpha}{2\pi n} - \frac{\varphi_n}{2\pi}}(u_1) V_{-\frac{2k}{n} - \frac{\alpha}{\pi n} + \frac{\varphi_n}{2\pi}}(v_1) V_{\frac{k}{n} + \frac{\alpha}{2\pi n}}(v_2) \Big\rangle. \end{aligned} \tag{78}$$

Using the correlation function in Eq. (60), the final result is [1]

$$Z_{R_1,k}^{(\nu)}(\alpha) = |\theta_1(r_1|\tau)|^{-2(\frac{k}{n} + \frac{\alpha}{2\pi n} - \frac{\varphi_n}{2\pi})(\frac{2k}{n} + \frac{\alpha}{\pi n} - \frac{\varphi_n}{2\pi})} |\theta_1(r_2|\tau)|^{-2(\frac{k}{n} + \frac{\alpha}{2\pi n})(\frac{2k}{n} + \frac{\alpha}{\pi n} - \frac{\varphi_n}{2\pi})}$$

$$|\theta_1(r_1 + r_2|\tau)|^{2(\frac{k}{n} + \frac{\alpha}{2\pi n})(\frac{k}{n} + \frac{\alpha}{2\pi n} - \frac{\varphi_n}{2\pi})} \times \Big|\frac{\epsilon}{L}\partial_z \theta_1(0|\tau)\Big|^{-\Delta_k(\alpha)} \Big|\frac{\theta_\nu((\frac{k}{n} + \frac{\alpha}{2\pi n})(r_2 - r_1) + \frac{\varphi_n}{2\pi}r_1|\tau)}{\theta_\nu(0|\tau)}\Big|^2, \tag{79}$$

---

[1] Differently from Eq. (41) in [85] or Eq. (80) in [81], rather then using the absolute values we explicitly change $\varphi_n \to \varphi_n - 2\pi$ for $k < 0$.

where

$$\Delta_k(\alpha) = -6\frac{k^2}{n^2} - 6\frac{k\alpha}{n^2\pi} - 3\frac{\alpha^2}{2n^2\pi^2} + 3k\frac{\varphi_n}{n\pi} + 3\frac{\alpha\varphi_n}{2n\pi^2} - \frac{\varphi_n^2}{2\pi^2} - 2\theta(-k)(1 + \frac{3k}{n} + \frac{3\alpha}{2n\pi} - \frac{\varphi_n}{\pi}), \quad (80)$$

and $\theta(x)$ is the step function. It is important to note that for $k < 0$, we have to modify the flux at $u_1$ and $v_1$, $\varphi_n$, by inserting an additional $2\pi$ and $-2\pi$ fluxes. Essentially, we need to find the dominant term with the lowest scaling dimension in the mode expansion, as discussed in Appendix B. Moreover, the case of odd $n = n_o$ requires particular attention: as $|\alpha| > 2/3\pi$, also the mode $k = 0$ requires an additional $2\pi$ and $-2\pi$ fluxes at $u_1$ and $v_1$, respectively. Putting together the various pieces and using Eq. (74), the logarithm of the charged moments of $\rho_A^{R_1}$ are given by

$$\log N_n^{(v)}(\alpha) = \log N_{n,0}(\alpha) + \log N_{n,1}^{(v)}(\alpha), \quad (81)$$

where the spin-independent part is

$$\log N_{n,0}(\alpha) = \log R_n - \frac{\alpha^2}{2\pi^2 n} \log\left|\theta_1(r_1|\tau)^2 \theta_1(r_2|\tau)^2 \theta(r_1 + r_2|\tau)^{-1}\left(\frac{\epsilon}{L}\partial_z\theta_1(0|\tau)\right)^{-3}\right|,$$

$$\log R_{n_o} = -\left(\frac{n_o^2 - 1}{12n_o}\right)\log\left|\theta_1(r_1|\tau)\theta_1(r_2|\tau)\theta(r_1 + r_2|\tau)\left(\frac{\epsilon}{L}\partial_z\theta_1(0|\tau)\right)^{-3}\right|,$$

$$\log R_{n_e} = -\left(\frac{n_e^2 - 4}{12n_e}\right)\log\left|\theta_1(r_1|\tau)\theta_1(r_2|\tau)\left(\frac{\epsilon}{L}\partial_z\theta_1(0|\tau)\right)^{-2}\right|$$

$$- \left(\frac{n_e^2 + 2}{12n_e}\right)\log\left|\theta(r_1 + r_2|\tau)\left(\frac{\epsilon}{L}\partial_z\theta_1(0|\tau)\right)^{-1}\right|. \quad (82)$$

The first equation for $N_{n,0}(\alpha)$ is always valid for any alpha for $n = n_e$ even, but only in the region $|\alpha| < 2/3\pi$ for $n = n_o$ odd; otherwise it must be modified as

$$\log N_{n_o,0}(\alpha) = \log R_{n_o} - \frac{\alpha^2}{2\pi^2 n_o}\log\left|\theta_1(r_1|\tau)^2\theta_1(r_2|\tau)^2\theta(r_1 + r_2|\tau)^{-1}\left(\frac{\epsilon}{L}\partial_z\theta_1(0|\tau)\right)^{-3}\right|$$

$$+ \frac{|\alpha|}{n_o\pi}\log\left|\theta_1(r_1|\tau)^3\theta_1(r_2|\tau)\theta(r_1 + r_2|\tau)^{-1}\left(\frac{\epsilon}{L}\partial_z\theta_1(0|\tau)\right)^{-3}\right|$$

$$- \frac{2}{n}\log\left|\theta_1(r_1|\tau)\left(\frac{\epsilon}{L}\partial_z\theta_1(0|\tau)\right)^{-1}\right|, \quad \text{for } |\alpha| > 2/3\pi. \quad (83)$$

Hence for odd $n = n_o$, the exponent of the charged moments $N_{n_o}^{(v)}(\alpha)$ has a discontinuity as a function of $\alpha$ for $|\alpha| = \frac{2\pi}{3}$. This singular behaviour in $\alpha$ is reminiscent of what was found for the negativity spectrum of free fermions in [86]. Let us also note that the above result does not hold for $n_o = 1$, for which we will provide an analytical expression in the following. The spin structure dependent term is

$$\log N_{n,1}^{(v)}(\alpha) = 2\sum_{k=-(n-1)/2}^{(n-1)/2}\log\left|\frac{\theta_v((\frac{k}{n} + \frac{\alpha}{2\pi n})(r_2 - r_1) + \frac{\varphi_n}{2\pi}r_1|\tau)}{\theta_v(0|\tau)}\right|. \quad (84)$$

Although our main focus is the state with $v = 3$, we notice that $N_{n,1}^{(1)}(\alpha)$ above is strictly infinite because $\theta_1(0|\tau) = 0$. This is related to the fermion zero mode in this sector and is not a prerogative of the charged quantities.

In the case of intervals of equal lengths $\ell_1 = \ell_2 = \ell$ the charged logarithmic negativity

(i.e., $\mathcal{E}^{\nu}(\alpha) \equiv \lim_{n_e \to 1} \log N_{n_e}^{(\nu)}(\alpha)$) simplifies as

$$\mathcal{E}^{\nu}(\alpha) = \mathcal{E}^{(\nu)} - \frac{\alpha^2}{2\pi^2} \log \left| \theta_1(r|\tau)^4 \theta(2r_1|\tau)^{-1} \left( \frac{\epsilon}{L} \partial_z \theta_1(0|\tau) \right)^{-3} \right|,$$

$$\text{with} \qquad \mathcal{E}^{(\nu)} = \frac{1}{4} \log \left| \theta_1(r|\tau)^2 \theta(2r|\tau)^{-1} \left( \frac{\epsilon}{L} \partial_z \theta_1(0|\tau) \right)^{-1} \right| + 2 \log \left| \frac{\theta_\nu(\frac{r}{2}|\tau)}{\theta_\nu(0|\tau)} \right|, \quad (85)$$

where $r = \ell/L$.

Eq. (85) represents our final field theoretical result for the charged logarithmic negativities in a tripartite geometry with two equal intervals. We now test this prediction against exact lattice computations obtained with the techniques reported in Appendix A. However, for a direct comparison without fitting parameters, we have to take into account the non-universal contribution coming from the discretisation of the spatial coordinate, i.e. the explicit expression for the cutoff $\epsilon$ in (85) that does depend also on $\alpha$, but not on the size and temperature. We can exploit the latter property to deduce its exact value from the knowledge of the lattice negativities at $T = 0$ in the thermodynamic limit that can be determined via Fisher-Hartwig techniques, as reported in Appendix C, cf. Eq. (179). The numerical results for the charged negativities are shown in Fig. 2, where four panels highlight the dependence on $\ell, T, \alpha$, and $\ell/L$, respectively. The agreement with the parameter-free asymptotic results (85) is always excellent. Let us critically discuss these results. First, it is known for $\alpha = 0$, the logarithmic negativity saturates at finite temperature once $\ell T \gg 1$ [85], i.e., obeys an area law; conversely the top-left panel of Fig. 2 shows that $\mathcal{E}(\alpha)$ follows a volume law. This scaling can be also inferred analytically from the high-temperature limit reported in the following subsection. In the top-right panel of the same figure, we observe that $\mathcal{E}(\alpha)$ has a plateau at low temperatures, i.e when $T \ll 1/L$ so that the temperature is smaller than the energy finite-size gap (of order $1/L$); consequently the system behaves as if it is at zero temperature with exponentially small corrections in $TL$. For larger $T$ a linear decrease sets up for low enough $T$, before an exponential high temperature behaviour takes place (this is not shown in the picture, but see next subsection). In the bottom-left panel of Fig. 2, we analyse the $\alpha$ dependence fixing $\ell$ and $L$ for a few values of $\beta$. We observe a fairly good agreement between lattice and field theory, although when $\alpha$ gets closer to $\pm\pi$ the agreement gets worse. This is not surprising because charged quantities exactly at $\pm\pi$ are known to be singular [98] and consequently finite $\ell$ effects are more severe. Moreover, the plot clearly shows that $\mathcal{E}(\alpha)$ has a differentiable maximum in $\alpha = 0$ (that we need for the saddle point approximation). In the bottom-right panel, we show that the difference $\mathcal{E}(\alpha, T) - \mathcal{E}(\alpha, 0)$ is a universal function of $\beta/L$ and $\ell/L$: we verify this behaviour by looking at various system sizes, $L$, and showing that they all collapse on the same curve. The agreement also slightly improves as $L$ increases, as it should.

Let us conclude this subsection reporting the result for the charged probability $N_1(\alpha) = \text{Tr}[\rho^{R_1} e^{i\hat{Q}\alpha}]$ that requires to specialise the above discussion to the case $n = 1$. Hence, $N_1^{(\nu)}(\alpha)$ reduces to one mode, $k = 0$, and Eq. (77) becomes

$$\frac{1}{2\pi} \epsilon^{\mu\nu} \partial_\nu A_\mu^0(x) = \left( \frac{\alpha}{2\pi} \right) \delta(x - u_1) - \left( \frac{\alpha}{\pi} \right) \delta(x - v_1) + \left( \frac{\alpha}{2\pi} \right) \delta(x - v_2). \qquad (86)$$

As detailed in the last part of Appendix B, we need to find the dominant term, i.e. with the lowest scaling dimension, in the mode expansion. In particular, it turns out that for $|\alpha/\pi| > 2/3$ an additional $-2\pi$ flux has to be inserted at $v_1$ while an additional $2\pi$ has to be added at $u_1$ or, equivalently, at $v_2$. This is the only difference with respect to $n_o \neq 1$, when the $2\pi$ flux has

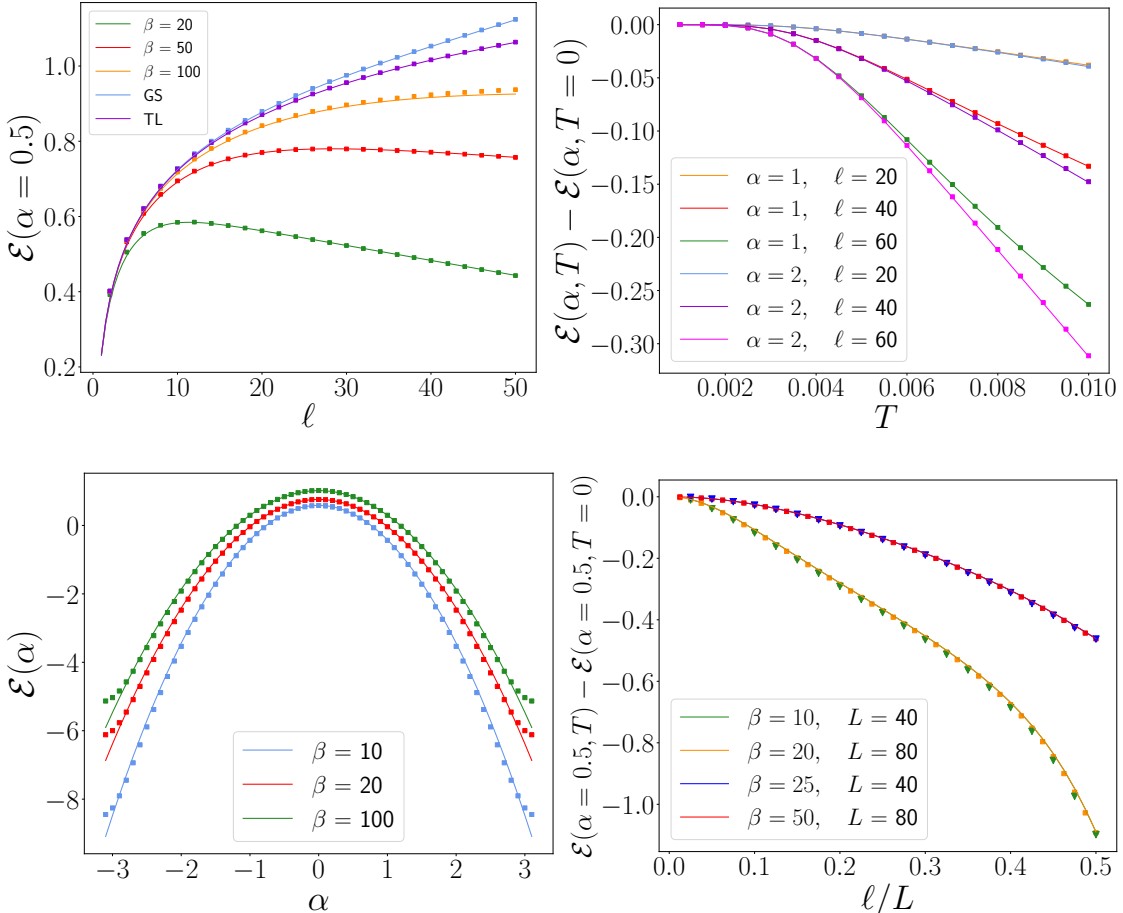

Figure 2: Charged negativity $\mathcal{E}(\alpha)$ in a tripartite torus with subsystem length $\ell_1 = \ell_2 = \ell$. CFT results (85), lines, against numerics on the lattice, symbols. Top-left: $\mathcal{E}(\alpha)$ as a function of $\ell$ for $\alpha = 0.5$. We consider different values of $\beta = 1/T$: in particular, GS stands for ground state, i.e. $T = 0$ while TL refers to the thermo-dynamic limit $T = 0, L \to \infty$. System size is fixed to $L = 200$ sites, except for the TL curve. Top-right: $\mathcal{E}(\alpha)$ as a function of the temperature $T$ for different values of $\alpha$ and $\ell$, with $L = 200$. The subtraction of the value $\mathcal{E}(\alpha, T = 0)$ cancels the dependence on the cutoff and the resulting curves are universal. Bottom-left: $\mathcal{E}(\alpha)$ as a function $\alpha$ for $L = 200$ and $\ell = 20$ for a few $\beta$. The agreement is perfect away from the boundaries $\alpha = \pm\pi$. Bottom-right: Scaling collapse of the charged negativity as a function of $\beta/L$ and $\ell/L$. We fix $\alpha = 0.5$.

to be inserted only in $u_1$. Hence, the final expression is given by

$$
N_1^{(\nu)}(\alpha) = \begin{cases} \dfrac{|\theta_1(r_1|\tau)|^{-\frac{\alpha^2}{\pi^2}} |\theta_1(r_2|\tau)|^{-\frac{\alpha^2}{\pi^2}} |\theta_1(r_1+r_2|\tau)|^{\frac{\alpha^2}{2\pi^2}}}{|\epsilon_N/L\partial_z\theta_1(0|\tau)|^{-\frac{3\alpha^2}{2\pi^2}}} \left| \dfrac{\theta_\nu(|\frac{\alpha}{2\pi}|(r_2-r_1)|\tau)}{\theta_\nu(0|\tau)} \right|^2 & |\alpha| \leq \frac{2\pi}{3} \\[3ex] f(r_1,r_2;|\alpha|) \dfrac{\theta_1(r_1+r_2|\tau)|^{\frac{\alpha}{\pi}|(|\frac{\alpha}{2\pi}|-1)}}{|\epsilon_N/L\partial_z\theta_1(0|\tau)|^{-\frac{3|\alpha|(-|\alpha|+2\pi)}{2\pi^2}-2}} \left| \dfrac{\theta_\nu(|\frac{\alpha}{2\pi}|(r_2-r_1)+r_1|\tau)}{\theta_\nu(0|\tau)} \right|^2 & |\alpha| > \frac{2\pi}{3} \end{cases}, \quad (87)
$$

where $f(x, y; q) = \frac{1}{2}[x^{2(q-1)(-2q+1)} y^{2q(-2q+1)} + x \leftrightarrow y]$. The cutoff related to the charged probability is denoted as $\epsilon_N$. Its explicit expression, for a lattice regularisation of the Dirac field, is given in Eq. (181a) and (181b) for $|\alpha| \leq 2/3\pi$ and $|\alpha| > 2/3\pi$, respectively. This introduction of a new symbol $\epsilon_N$ is necessary in order to avoid confusion with the cutoff $\epsilon$

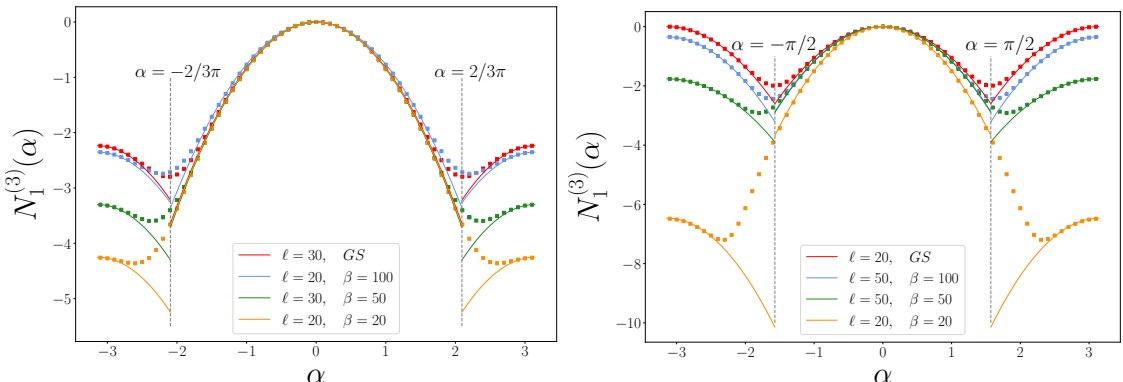

Figure 3: The charged probability $N_1^{(3)}(\alpha)$ for tripartite (left) and bipartite (right) geometry as a function of $\alpha$. We set $L = 100$. Analytical predictions in Eqs. (87) and (119) are compared with the exact lattice computations at different $\beta$. Notice the discontinuities at $\alpha = \pm 2/3\pi$ (left) and $\alpha = \pm \pi/2$ (right).

obtained in the replica limit as $n_e \to 1$, given explicitly for the lattice model in Eq. (180) (and it is different from $n_o = 1$).

## 4.1 Low and high temperature limits.

In this section we report the low and high temperature limits of the charged Rényi negativity. Actually, the results that we are going to derive in the following for the tripartite geometry can be much more easily deduced by mapping the results in the complex plane (29) (i.e. both $L, \beta \to \infty$) to a cylinder periodic in either space or time (obtaining the forthcoming Eqs. (92) and (99), respectively). It is however a highly non trivial check for the correctness of our formulas that these results are re-obtained in the proper limits. For sake of conciseness, we focus on even $n = n_e$ and on the $\nu = 3$ sector, but similar formulas hold for all other cases.

In the low temperature limit where $\tau = i\beta/L \to i\infty$, we can take advantage of the relation

$$\lim_{\beta \to \infty} \theta_1(z|i\beta/L) = 2e^{-\pi\beta/(4L)} \sin \pi z + O(e^{-2\pi\beta/L}). \tag{88}$$

In this way we obtain for the spin-independent part

$$\log N_{n_e,0}(\alpha) = \log R_{n_e} - \frac{\alpha^2}{2\pi^2 n_e} \ln \left| \left(\frac{L}{\pi\epsilon}\right)^3 \frac{\sin^2\left(\frac{\pi\ell_1}{L}\right)\sin^2\left(\frac{\pi\ell_2}{L}\right)}{\sin\left(\frac{\pi(\ell_1+\ell_2)}{L}\right)} \right| + O(e^{-2\pi/(LT)}), \tag{89}$$

while using the product representation of the theta function (65), the spin structure dependent term (84) can be rewritten as

$$\log N_{n_e,1}^{(3)}(\alpha) = 2\sum_{j=1}^{\infty} \frac{(-1)^{j+1}}{j} \frac{1}{\sinh(\pi j\beta/L)} \left( \cos(j(r_1-r_2)\alpha/n_e) \frac{\sin(\pi j r_2) - \sin(\pi j r_1)}{\sin(\pi j(r_2-r_1)/n_e)} - n_e \right). \tag{90}$$

Thus, at the leading order, Eq. (89) is the whole story at zero temperature, since in the replica limit the above expression contributes to the charged negativity as

$$\mathcal{E}_1^{(3)}(\alpha) = \lim_{n_e \to 1} \log N_{n_e,1}^{(3)}(\alpha) = 4e^{-\pi/(LT)} \left( \cos((r_1-r_2)\alpha) \frac{\cos(\pi(r_2+r_1)/2)}{\cos(\pi(r_2-r_1)/2)} - 1 \right). \tag{91}$$

Putting everything together, in the low temperature limit the logarithmic charged negativity of two adjacent intervals for spatially antiperiodic fermions is given by

$$\mathcal{E}(\alpha, LT \ll 1) = \mathcal{E} - \frac{\alpha^2}{2\pi^2 n_e} \log \left| \left( \frac{L}{\pi\epsilon} \right)^3 \frac{\sin^2\left(\frac{\pi\ell_1}{L}\right)\sin^2\left(\frac{\pi\ell_2}{L}\right)}{\sin\left(\frac{\pi(\ell_1+\ell_2)}{L}\right)} \right| + O(e^{-2\pi/(LT)}), \qquad (92)$$

where $\mathcal{E}(LT \ll 1) = \frac{1}{4}\log\left|\left(\frac{L}{\pi\epsilon}\right)\frac{\sin(\frac{\pi\ell_1}{L})\sin(\frac{\pi\ell_2}{L})}{\sin(\frac{\pi(\ell_1+\ell_2)}{L})}\right|$. We can also study the low-temperature behaviour of Eq. (87), which reads

$$N_1(\alpha, LT \ll 1) \simeq$$
$$\begin{cases} -\frac{\alpha^2}{2\pi^2}\log\left|\frac{L^3}{\pi^3\epsilon_N^3}\frac{\sin^2(\frac{\pi\ell_1}{L})\sin^2(\frac{\pi\ell_2}{L})}{\sin(\frac{\pi(\ell_1+\ell_2)}{L})}\right|, & |\alpha| \le \frac{2\pi}{3} \\ \frac{(2\pi-|\alpha|)|\alpha|}{2\pi^2}\log\left|\frac{L^3}{\pi^3\epsilon_N^3}\frac{\sin^2(\frac{\pi\ell_1}{L})\sin^2(\frac{\pi\ell_2}{L})}{\sin(\frac{\pi(\ell_1+\ell_2)}{L})}\right| - \log\left|\frac{L^2}{\pi^2\epsilon_N^2}\sin(\frac{\pi\ell_1}{L})\sin(\frac{\pi\ell_2}{L})\right|. & |\alpha| > \frac{2\pi}{3}. \end{cases} \qquad (93)$$

To investigate the high temperature behaviour, $\tau = i\beta/L \to 0$, we can use the modular transformation rules for the theta functions:

$$\begin{aligned} \theta_1(z|\tau) &= -(-i\tau)^{-1/2}e^{-i\pi z^2/\tau}\theta_1(z/\tau|-1/\tau), \\ \theta_3(z|\tau) &= (-i\tau)^{-1/2}e^{-i\pi z^2/\tau}\theta_3(z/\tau|-1/\tau), \end{aligned} \qquad (94)$$

and the asymptotic form of the $\theta_1$ function in the small $\beta$ limit

$$\theta_1(z/\tau|-1/\tau) = -2ie^{-\frac{\pi L}{4\beta}}\sinh(\frac{\pi z L}{\beta}) + O(e^{\frac{3\pi L}{\beta}(z-3/4)}), \quad 0 \le z \le 1/2. \qquad (95)$$

Therefore, the leading terms of the spin-independent part of the charged negativities can be written as

$$\log N_{n_e,0}(\alpha) = \log R_{n_e} + \frac{(\ell_1-\ell_2)^2\alpha^2}{2\pi n_e \beta L} - \frac{\alpha^2}{2\pi^2 n_e}\ln\left|\left(\frac{\beta}{\pi\epsilon}\right)^3\frac{\sinh^2\left(\frac{\pi\ell_1}{\beta}\right)\sinh^2\left(\frac{\pi\ell_2}{\beta}\right)}{\sinh\left(\frac{\pi(\ell_1+\ell_2)}{\beta}\right)}\right| + O(e^{-\pi LT}), \qquad (96)$$

while for the spin structure dependent term (84) we find

$$\log N_{n_e,1}^{(3)}(\alpha) = -\frac{\pi}{2\beta L}\left[\left(\frac{n_e^2-1}{3n_e}\right)(\ell_2-\ell_1)^2 + n_e\ell_1(\ell_2-\ell_1) + n_e\ell_1^2\right] - \frac{(\ell_2-\ell_1)^2\alpha^2}{2\pi L\beta n_e} +$$
$$-2\sum_{j=1}^{\infty}\frac{(-1)^j}{j}\frac{1}{\sinh(\frac{\pi j L}{\beta})}\left(\cosh\left(\frac{j(\ell_1-\ell_2)\alpha}{\beta n_e}\right)\frac{\sinh(\pi\ell_2 j/\beta) - \sinh(\pi\ell_1 j/\beta)}{\sinh\left(\frac{\pi(\ell_2-\ell_1)j}{n_e\beta}\right)} - n_e\right). \qquad (97)$$

For fixed $\ell_{1,2}/\beta$ and $\tau = i\beta/L \to 0$ we get

$$\mathcal{E}_1^{(3)}(\alpha) = -\frac{\pi\ell_1\ell_2}{2\beta L} - \frac{(\ell_2-\ell_1)^2\alpha^2}{2\pi L\beta}, \qquad (98)$$

and therefore,

$$\mathcal{E}(\alpha, LT \gg 1) = \mathcal{E} - \frac{\alpha^2}{2\pi^2 n_e}\log\left|\left(\frac{\beta}{\pi\epsilon}\right)^3\frac{\sinh^2\left(\frac{\pi\ell_1}{\beta}\right)\sinh^2\left(\frac{\pi\ell_2}{\beta}\right)}{\sinh\left(\frac{\pi(\ell_1+\ell_2)}{\beta}\right)}\right| + O(e^{-\pi LT}), \qquad (99)$$

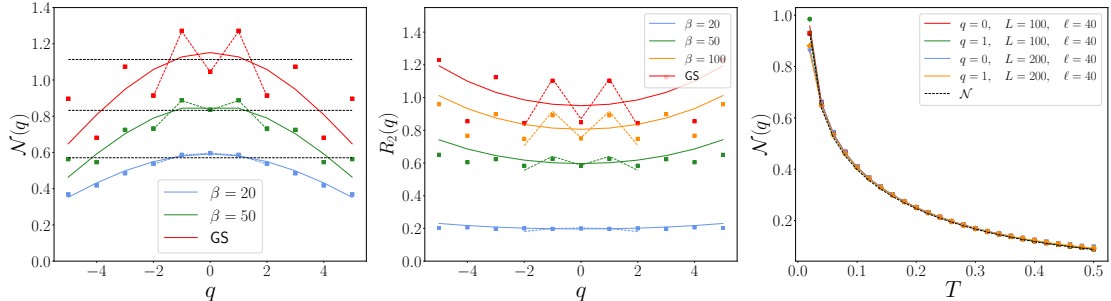

Figure 4: Imbalance resolved negativities for a few different values of $q$, $L = 200$, $\ell_1 = \ell_2 = \ell = 30$, with $n_e \to 1$ (left-panel) and $n = 2$ (middle-panel). The dashed black lines are the truly asymptotic result (108) showing equipartition, while the solid lines include the first correction due to the cutoffs as in Eq. (107). The dashed coloured lines are the ratio between the Fourier transforms without exploiting the saddle point approximation. For small $q$, the field theory prediction (in which the lattice cutoffs are included) well describes the numerical data. In the right-panel, $\ell = 40$ is fixed, we report two system sizes and two values of $q$ and plot $\mathcal{N}(q)$ as a function of $T$. The coloured lines are Eq. (107) while the dashed one represents Eq. (108). The plot confirms the equipartition of negativity. Moreover, for large $T$, $\mathcal{N}(q)$ becomes a universal function of $\pi \ell T$.

where $\mathcal{E}(LT \gg 1) = \frac{1}{4}\log\left|\left(\frac{\beta}{\pi\epsilon}\right)\frac{\sinh(\frac{\pi\ell_1}{\beta})\sinh(\frac{\pi\ell_2}{\beta})}{\sinh(\frac{\pi(\ell_1+\ell_2)}{\beta})}\right|$. This limit confirms analytically the volume law behaviour observed in Fig. 2.

The high-temperature limit of the charged probability $N_1(\alpha)$ in Eq. (87) is

$$N_1(\alpha, LT \gg 1) \simeq$$
$$\begin{cases} -\frac{\alpha^2}{2\pi^2}\log\left|\frac{\beta^3}{\pi^3\epsilon_N^3}\frac{\sinh^2(\frac{\pi\ell_1}{L})\sinh^2(\frac{\pi\ell_2}{L})}{\sinh(\frac{\pi(\ell_1+\ell_2)}{L})}\right| & |\alpha| \leq \frac{2\pi}{3}, \\ \frac{(2\pi-|\alpha|)|\alpha|}{2\pi^2}\log\left|\frac{\beta^3}{\pi^3\epsilon_N^3}\frac{\sinh^2(\frac{\pi\ell_1}{L})\sinh^2(\frac{\pi\ell_2}{L})}{\sinh(\frac{\pi(\ell_1+\ell_2)}{L})}\right| - \log\left|\frac{\beta^2}{\pi^2\epsilon_N^2}\sinh(\frac{\pi\ell_1}{L})\sinh(\frac{\pi\ell_2}{L})\right|. & |\alpha| > \frac{2\pi}{3} \end{cases} \tag{100}$$

Let us conclude the subsection comparing these new results with those for the standard (bosonic) charged negativity reported in Eq. (29). At zero temperature and in the thermodynamic limit $\ell_i \ll L$, Eq. (89) matches exactly the bosonic negativity (29) (at $K = 1$ to describe free fermions) obtained in the same limit. As discussed deeply in Ref. [85] for the Rényi negativity (at $\alpha = 0$), this shows that the choice of charged moments of the partial TR we made in Eq. (72) provides a partition function evaluated on the same worldsheet $\mathcal{R}_{n,\alpha}$ as the one for the moments of the standard charged partial transpose in [91].

## 4.2 Symmetry resolution

Again for conciseness of the various formulas, in this subsection we focus on the case $\ell_1 = \ell_2 = \ell$ (when also a closed-form expression for the spin-dependent part is easier to write), but more general formulas are similarly derived. Since we are ultimately using a saddle point approximation to make the Fourier transform (75), the charged moments (72) can be truncated at Gaussian level in $\alpha$ as

$$N_n^{(\nu)}(\alpha) = R_n^{(\nu)} e^{-b_n \alpha^2/2}, \tag{101}$$

where

$$b_n = \frac{1}{\pi^2 n}\log\left|\theta_1(r_1|\tau)^4\theta(2r_1|\tau)^{-1}\left(\frac{\epsilon}{L}\partial_z\theta_1(0|\tau)\right)^{-3}\right|. \tag{102}$$

The Fourier transform reads

$$\mathcal{Z}_{R_1,n}^{(\nu)}(q) = R_n^{(\nu)} \int_{-\pi}^{\pi} \frac{d\alpha}{2\pi} e^{-iq\alpha} e^{-\alpha^2 b_n/2}, \tag{103}$$

where we used that the expectation value of the charge imbalance operator $\hat{Q}_A$ for a free Dirac field is $\bar{q} = 0$ at any temperature. In the saddle point approximation the integration domain is extended to the whole real line and we end up in a simple Gaussian integral, obtaining

$$\mathcal{Z}_{R_1,n}^{(\nu)}(q) \simeq \frac{R_n^{(\nu)}}{\sqrt{2\pi b_n}} e^{-\frac{q^2}{2b_n}}. \tag{104}$$

Through a similar analysis, we compute

$$p^{(\nu)}(q) = \int_{-\pi}^{\pi} \frac{d\alpha}{2\pi} e^{-iq\alpha} N_1^{(\nu)}(\alpha), \tag{105}$$

which through the saddle-point approximation reads

$$p(q) \simeq \frac{e^{-\frac{q^2}{2b_N}}}{\sqrt{2\pi b_N}} \qquad b_N = \frac{1}{\pi^2} \log \left| \theta_1(r_1|\tau)^4 \theta(2r_1|\tau)^{-1} \left( \frac{\epsilon_N}{L} \partial_z \theta_1(0|\tau) \right)^{-3} \right|. \tag{106}$$

Let us note that for $\alpha \in [-\pi, \pi]$, the quantity $N_1^\nu(\alpha)$ has a global maximum for $\alpha = 0$ and two local maxima for $\alpha = \pm\pi$, see Fig. 3 (left). However, since $N_1^\nu(\pm\pi) < N_1^\nu(0)$, we can neglect the contributions to the integral coming from the regions close to the extrema at $\alpha = \pm\pi$. A similar reasoning applies to all odd charged moments $R_{n_o}(\alpha)$. Once again, let us stress the difference between the cutoff $\epsilon_N$ and the cutoff $\epsilon$ obtained in the replica limit $b = \lim_{n_e \to 1} b_{n_e}$, whose lattice expression is given in Eqs. (181a) and (180), respectively. Putting everything together, we obtain

$$R_n^{(\nu)}(q) = R_n^{(\nu)} \sqrt{\frac{(2\pi b_N)^n}{2\pi b_n}} e^{-\frac{q^2}{2}\left(\frac{1}{b_n} - \frac{n}{b_N}\right)}, \qquad \mathcal{N}^{(\nu)}(q) = \frac{1}{2}\left( e^{\mathcal{E}^{(\nu)}} \sqrt{\frac{b_N}{b}} e^{-\frac{q^2}{2}\left(\frac{1}{b} - \frac{1}{b_N}\right)} - 1 \right). \tag{107}$$

When the $O(1)$ terms are negligible with respect to the leading order ones in the variance, $b_N \simeq b$, hence

$$\mathcal{N}^{(\nu)}(q) \simeq \mathcal{N}^{(\nu)}, \tag{108}$$

i.e. exact equipartition of negativity in the different imbalance sectors at leading order, as shown for the bosonic negativity in section (2.2). A similar result holds even if $\ell_1 \neq \ell_2$ in the low/high temperature limits, it would be sufficient to modify the expression of the variances in Eqs. (102) and (106).

It is instructive to explicitly write down the first term breaking the equipartition. For large $L$, we can expand the exponential in Eq. (107) as

$$e^{-\frac{q^2}{2}\left(\frac{1}{b} - \frac{1}{b_N}\right)} \simeq 1 - \frac{q^2 \log(|\epsilon/\epsilon_N|)\pi^2}{6(\log L)^2} \equiv 1 - \frac{\gamma}{(\log L)^2} q^2, \tag{109}$$

and

$$\sqrt{\frac{b_N}{b}} \simeq 1 + \frac{\gamma'}{\log L}, \tag{110}$$

where $\gamma$ and $\gamma'$ are implicitly defined, also in terms of the cutoffs $\epsilon$ and $\epsilon_N$ in Appendix C. To sum up, we get

$$\mathcal{N}^{(\nu)}(q) \simeq \mathcal{N}^{(\nu)}\left( 1 + \frac{\gamma'}{\log L} - \frac{\gamma}{(\log L)^2} q^2 + \dots \right), \tag{111}$$

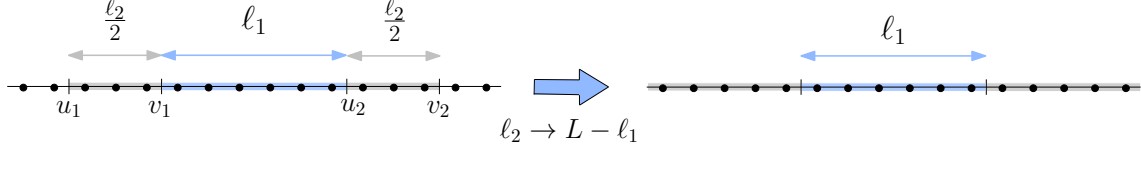

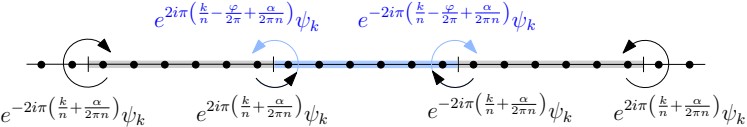

Figure 5: The tripartite geometry (left) of an interval of length $\ell_1$ symmetrically embedded inside another subsystem of total length $\ell_2$. A single interval in a chain of total length $L$ (right) is obtained taking the limit $\ell_2 \to L - \ell_1$ of this tripartite geometry. The bottom panel shows the phase taken by the field $\psi_k$ going around each branch point. The reduced density matrix corresponds to the union of the coloured regions, and the partial transpose is applied to the blue region.

where we have derived the leading $q$-dependent contributions and shown that the equipartition is broken at order $1/(\log L)^2$.

In Fig. 4 we test the accuracy of our predictions against exact lattice numerical calculations. It is evident that equipartition is broken for all the values of $\ell, T, L$ we considered and the effect is more pronounced as $|q|$ is increased. However, the main smooth part of corrections to the scaling is captured by Eq. (107), see the full line in the plots, and does not come as a surprise. Also the presence of further subleading oscillating (in $q$) corrections have been observed for the resolved entropies [96] and were expected. In our case, such corrections are enhanced by the presence of the maxima at $\alpha = \pm\pi$ in $N_1(\alpha)$, see Fig. 4, that provide large corrections to the scaling in $p(q)$. Indeed, taking the Fourier transforms without making the saddle-point approximation, the agreement between numerics and field theory is perfect. As $\ell \gg 1/T$, all these corrections become smaller and imbalance resolved negativity flattens in $q$, mainly as a consequence of the lowering of the maxima at $\alpha = \pm\pi$ in $N_1(\alpha)$, see Fig. 4.

# 5 Charged and symmetry resolved negativities in a bipartite geometry

In this section we move to the imbalance resolved negativity of a single interval at finite temperature. This geometry can be studied more effectively by considering a tripartite geometry where an interval of length $\ell_1$ is symmetrically embedded inside another subsystem of total length $\ell_2$, as depicted in Fig 5. Eventually, we take the limit $\ell_2 \to L - \ell_1$ in our calculations, where $L$ is the total length of the chain, in such a way that the part $B$ becomes the empty set and consequently the system becomes bipartite.

Choosing the locations of the branch points at $u_1 = -\ell_2/(2L) = -r_2/2, v_2 = (\ell_2/2 + \ell_1)/L = r_2/2 + r_1, u_2 = \ell_1/L = r_1, v_1 = 0$, the multivalued field $\psi_k$ takes up a phase $e^{2\pi i(\frac{k}{n} + \frac{\alpha}{2\pi n} - \frac{\varphi_n}{2\pi})}$, $e^{-2\pi i(\frac{k}{n} + \frac{\alpha}{2\pi n} - \frac{\varphi_n}{2\pi})}$ going around $v_1$ and $u_2$, respectively while going around $v_2$ and $u_1$ picks up a phase $e^{2\pi i(\frac{k}{n} + \frac{\alpha}{2\pi n})}$, $e^{-2\pi i(\frac{k}{n} + \frac{\alpha}{2\pi n})}$, respectively. As repeatedly used, this multivaluedness of the field $\psi_k$ can be removed by introducing a single-valued field coupled to an external gauge field $A_\mu^k$, as in Eq. (53). In order to recover the correct phases of the multivalued fields, $A_\mu^k$ has to

satisfy

$$\frac{1}{2\pi}\epsilon^{\mu\nu}\partial_\nu A_\mu^k(x) = \left(\frac{2k}{n}+\frac{\alpha}{\pi n}-\frac{\varphi_n}{2\pi}\right)(\delta(x-v_1)-\delta(x-u_2))+\left(\frac{k}{n}+\frac{\alpha}{2\pi n}\right)(\delta(x-v_2)-\delta(x-u_1)).$$
(112)

Therefore, $Z_{R_1,k}^{(\nu)}(\alpha)$ can be expressed as the following correlation function of vertex operators

$$Z_{R_1,k}^{(\nu)}(\alpha) = \left\langle V_{-\frac{k}{n}-\frac{\alpha}{2\pi n}}(u_1)V_{\frac{2k}{n}+\frac{\alpha}{\pi n}-\frac{\varphi_n}{2\pi}}(v_1)V_{-\frac{2k}{n}-\frac{\alpha}{\pi n}+\frac{\varphi_n}{2\pi}}(u_2)V_{\frac{k}{n}+\frac{\alpha}{2\pi n}}(v_2)\right\rangle.$$
(113)

Using Eq. (60), we have

$$Z_{R_1,k}^{(\nu)}(\alpha) = |\theta_1(r_1|\tau)|^{-2\left(\frac{2k}{n}+\frac{\alpha}{\pi n}-\frac{\varphi_n}{2\pi}\right)^2}\left|\frac{\theta_1(\frac{r_2}{2}|\tau)}{\theta_1(\frac{r_2}{2}+r_1|\tau)}\right|^{-4\left(\frac{k}{n}+\frac{\alpha}{2\pi n}\right)\left(\frac{2k}{n}+\frac{\alpha}{\pi n}-\frac{\varphi_n}{2\pi}\right)}$$

$$|\theta_1(r_1+r_2|\tau)|^{-2\left(\frac{k}{n}+\frac{\alpha}{2\pi n}\right)^2}\times\left|\frac{\epsilon}{L}\partial_z\theta_1(0|\tau)\right|^{-\Delta_k(\alpha)}\left|\frac{\theta_\nu\left(\left(\frac{k}{n}+\frac{\alpha}{2\pi n}\right)(r_2-r_1)+\frac{\varphi_n}{2\pi}r_1|\tau\right)}{\theta_\nu(0|\tau)}\right|,\quad (114)$$

where

$$\Delta_k(\alpha) = -10\frac{k^2}{n^2}-10\frac{k\alpha}{n^2\pi}-5\frac{\alpha^2}{2n^2\pi^2}+4k\frac{\varphi_n}{n\pi}+2\frac{\alpha\varphi_n}{n\pi^2}-\frac{\varphi_n^2}{2\pi^2}-2\theta(-k)(1+\frac{4k}{n}+\frac{2\alpha}{n\pi}-\frac{\varphi_n}{\pi}).\ (115)$$

Also in this case we fix the value of $\varphi_n$ for $k < 0$ according to the discussion in Appendix B, taking care of the mode $k = 0$ for $n = n_o$. This leads to the spin-independent terms

$$\log N_{n,0}(\alpha) = \log R_n - \frac{\alpha^2}{2\pi^2 n}\log\left|\frac{\theta_1(r_1|\tau)^4\theta_1\left(\frac{r_2}{2}|\tau\right)^4\theta_1(r_1+r_2|\tau)}{\theta_1\left(r_1+\frac{r_2}{2}|\tau\right)^4(\frac{\epsilon}{L}\partial_z\theta_1(0|\tau))^5}\right|,$$

$$\log N_{n_o,0}(|\alpha|>\pi/2) = \log R_{n_o} - \frac{\alpha^2}{2\pi^2 n_o}\log\left|\frac{\theta_1(r_1|\tau)^4\theta_1\left(\frac{r_2}{2}|\tau\right)^4\theta_1(r_1+r_2|\tau)}{\theta_1\left(r_1+\frac{r_2}{2}|\tau\right)^4(\frac{\epsilon}{L}\partial_z\theta_1(0|\tau))^5}\right|$$

$$+\frac{2|\alpha|}{\pi n_o}\log\left|\frac{\theta_1(r_1|\tau)^2\theta_1\left(\frac{r_2}{2}|\tau\right)}{\theta_1\left(r_1+\frac{r_2}{2}|\tau\right)(\epsilon/L\partial_z\theta_1(0|\tau))^2}\right|-\frac{2}{n}\log\left|\frac{\theta_1(r_1|\tau)}{(\frac{\epsilon}{L}\partial_z\theta_1(0|\tau))}\right|,$$

$$\log R_{n_o} = -\left(\frac{n_o^2-1}{6n_o}\right)\log\left|\frac{\theta_1(r_1|\tau)\theta_1\left(\frac{r_2}{2}|\tau\right)\theta_1(r_1+r_2|\tau)}{\theta_1\left(r_1+\frac{r_2}{2}|\tau\right)(\frac{\epsilon}{L}\partial_z\theta_1(0|\tau))^2}\right|,$$

$$\log R_{n_e} = -\left(\frac{n_e^2-4}{6n_e}\right)\log\left|\frac{\theta_1(r_1|\tau)\theta_1\left(\frac{r_2}{2}|\tau\right)}{\theta_1\left(r_1+\frac{r_2}{2}|\tau\right)(\frac{\epsilon}{L}\partial_z\theta_1(0|\tau))}\right|$$

$$-\left(\frac{n_e^2-1}{6n_e}\right)\log\left|\theta_1(r_1+r_2|\tau)(\frac{\epsilon}{L}\partial_z\theta_1(0|\tau))^{-1}\right|,$$

(116)

and

$$\log N_{n,1}^{(\nu)}(\alpha) = 2\sum_{k=-(n-1)/2}^{(n-1)/2}\log\left|\frac{\theta_\nu\left(\left(\frac{k}{n}+\frac{\alpha}{2\pi n}\right)(r_2-r_1)+\frac{\varphi_n}{2\pi}r_1|\tau\right)}{\theta_\nu(0|\tau)}\right|$$
(117)

for the spin structure dependent term. In this geometry, $N_{n_o}^{(\nu)}(\alpha)$ presents a discontinuity for $|\alpha| = \frac{\pi}{2}$, as shown for $n_o = 1$ in Fig. 3.

At this point, we derived all the needed formulas to take the limit $r_2 \to 1 - r_1$ and to reproduce the bipartite geometry in which we are interested. Using that $\theta_1(z+1) = \theta_1(z)$,

the spin-independent part of the charged logarithmic negativity (for even $n$ and for odd and $\alpha < |\pi/2|$) becomes

$$\log N_{n,0}(\alpha) = \log R_n - \frac{2\alpha^2}{\pi^2 n} \log \left| \frac{L\theta_1(r_1|\tau)}{\epsilon \partial_z \theta_1(0|\tau)} \right|, \tag{118}$$

while the spin-dependent ones are just given by (117) without major simplifications.

At this point, we analyse the charged probability $N_1(\alpha) = \text{Tr}[\rho^{R_1} e^{i\hat{Q}\alpha}]$. The final expression can be read off from Eq. (116) and, after some standard manipulations, can be put in the form

$$N_1^{(\nu)}(\alpha) = \begin{cases} \left( \frac{\theta_1(r_1|\tau)}{\frac{\epsilon_N}{L} \partial_z \theta_1(0|\tau)} \right)^{-2(\frac{\alpha}{\pi})^2} \left| \frac{\theta_\nu(|\frac{\alpha}{2\pi}|(1-2r_1)|\tau)}{\theta_\nu(0|\tau)} \right|^2 & |\alpha| \leq \frac{\pi}{2}, \\ \left( \frac{\theta_1(r_1|\tau)}{\frac{\epsilon_N}{L} \partial_z \theta_1(0|\tau)} \right)^{-2(|\frac{\alpha}{\pi}|-1)^2} \left| \frac{\theta_\nu(|\frac{\alpha}{2\pi}|(1-2r_1)+r_1|\tau)}{\theta_\nu(0|\tau)} \right|^2, & |\alpha| > \frac{\pi}{2}. \end{cases} \tag{119}$$

The cutoff for the charged probability is denoted by $\epsilon_N$ and its explicit expression is given in Eq. (183a) and (183b) for $|\alpha| \leq \pi/2$ and $|\alpha| > \pi/2$, respectively. We recall that, as in the tripartite case, $\epsilon_N$ is different from the cutoff $\epsilon$ obtained in the replica limit as $n_e \to 1$ and explicitly given in Eq. (182).

## 5.1 Low and high temperature limits

In this section we report the low and high temperature limits of the charged Rényi negativity, focussing, once again, on even $n = n_e$ and on the $\nu = 3$ sector. The low temperature limits of the spin-independent part, Eq. (116), can be obtained through the relation (88), finding

$$\log N_{n_e,0}(\alpha) = \log R_{n_e} - \frac{2\alpha^2}{\pi^2 n_e} \Big[ \log \Big| \frac{L}{\pi\epsilon} \sin\Big(\frac{\pi\ell_1}{L}\Big) \Big| \Big] + O(e^{-2\pi/(LT)}), \tag{120}$$

while the low temperature limit of the spin-dependent term, Eq. (117), can be obtained through the product representation (65) of the theta functions and it reads

$$\mathcal{E}_1^{(3)}(\alpha) = \lim_{n_e \to 1} \log Z_{n_e,1}^{(3)}(\alpha) = 4e^{-\pi/(LT)} \left( \cos((r_1 - r_2)\alpha) \frac{\cos(\pi(r_2 + r_1)/2))}{\cos(\pi(r_2 - r_1)/2)} - 1 \right). \tag{121}$$

Therefore, as $\tau = i\beta/L \to 0$ and $r_2 \to 1 - r_1$, we get in the replica limit

$$\mathcal{E}(LT \ll 1)(\alpha) = \Big( \frac{1}{2} - \frac{2\alpha^2}{\pi^2} \Big) \log \Big| \frac{L}{\pi\epsilon} \sin\Big(\frac{\pi\ell_1}{L}\Big) \Big| + O(e^{-2\pi/(LT)}). \tag{122}$$

We notice that Eq. (122) coincides with $2\log\text{Tr}(\rho_{A_1}^{1/2} e^{i\hat{Q}_{A_1}\alpha})$, as it should for pure states (and mentioned in the introduction for $\alpha = 0$).

The low-temperature limit of Eq. (119) is

$$\log N_1(\alpha) \simeq \begin{cases} -\frac{2\alpha^2}{\pi^2} \log \Big| \frac{L}{\pi\epsilon_N} \sin\big(\frac{2\pi\ell_1}{L}\big) \Big|, & |\alpha| \leq \frac{\pi}{2}, \\ -\frac{2(|\alpha|-\pi)^2}{\pi^2} \log \Big| \frac{L}{\pi\epsilon_N} \sin\big(\frac{2\pi\ell_1}{L}\big) \Big|, & |\alpha| > \frac{\pi}{2}. \end{cases} \tag{123}$$

Interestingly, the previous expansion shows that its Fourier transform vanishes for odd values of the imbalance $q$. This agrees with the discussion at the end of Section 2.3: as $\tau = i\beta/L \to 0$, the state becomes pure and the vanishing of $p(q)$ occurs because the parity of the imbalance is fixed by the conservation of $\hat{N}_1 + \hat{N}_2$. This reflects the fact that the entanglement is better resolved in subsystem charges rather than in the imbalance.

As done in Sec. 4.1, the high temperature limit can be obtained using the modular properties of theta functions, Eq. (94), and the relation (95). The result for the spin-independent part in Eq. (116) is

$$\log N_{n_e,0}(\alpha) = \log R_{n_e} - \frac{2\alpha^2}{\pi^2 n_e}\Big[-\frac{\pi^2\ell_1^2}{\beta L} + \log\Big|\frac{\beta}{\epsilon\pi}\sinh\Big(\frac{\pi\ell_1}{\beta}\Big)\Big|\Big] + O(e^{-\pi LT}). \tag{124}$$

The spin-dependent term in Eq. (116) can be evaluated as follows

$$\log N_{n_e,1}^{(3)}(\alpha) = -\frac{\pi}{2\beta L}\Big[\Big(\frac{n_e^2-1}{3n_e}\Big)(L-2\ell_1)^2 + n_e\ell_1(L-2\ell_1) + n_e\ell_1^2\Big] - \frac{(L-2\ell_1)^2\alpha^2}{2\pi L\beta n_e} +$$
$$-2\sum_{j=1}^{\infty}\frac{(-1)^j}{j}\frac{1}{\sinh(\frac{\pi jL}{\beta})}\Big(\cosh\Big(\frac{j(L-2\ell_1)\alpha}{\beta n_e}\Big)\frac{\sinh(\pi(L-\ell_1)j/\beta)-\sinh(\varphi_{n_e}\ell_1 j/\beta)}{\sinh\Big(\frac{\pi(L-2\ell_1)j}{n_e\beta}\Big)} - n_e\Big), \tag{125}$$

which gives in the replica limit, for $\beta/L \to 0$ and $\ell_1/\beta$ fixed,

$$\mathcal{E}_1^{(3)}(\alpha) = \frac{\pi\ell_1(\ell_1-L)}{2\beta L} - \frac{(L-2\ell_1)^2\alpha^2}{2\pi\beta L} + O(e^{-\pi LT}). \tag{126}$$

To sum up, we find in the high temperature regime

$$\mathcal{E}(LT\gg 1)(\alpha) = \Big(\frac{1}{2} - \frac{2\alpha^2}{\pi^2}\Big)\Big[\log\Big|\frac{\beta}{\pi\epsilon}\sinh(\frac{\pi\ell_1}{\beta})\Big| - \frac{\pi\ell_1}{\beta}\Big] - \frac{\alpha^2 L}{2\pi\beta} + O(e^{-\pi LT}). \tag{127}$$

Given the result found in Eq. (122), one could be tempted to do a conformal mapping to a cylinder periodic in time, to get

$$\mathcal{E}(LT\gg 1)(\alpha)_{\text{naive}} = \Big(\frac{1}{2} - \frac{2\alpha^2}{\pi^2}\Big)\log\Big|\frac{\beta}{\epsilon\pi}\sinh\Big(\frac{\pi\ell_1}{\beta}\Big)\Big|, \tag{128}$$

which is nothing but the finite temperature logarithmic charged entropy of order $1/2$. This naive derivation provides a wrong result whose origin has been extensively discussed in [46] and it remains the right interpretation also for $\alpha \neq 0$. Indeed, for pure states (i.e. $T \to 0$), the $n_e$–sheeted Riemann surface, $\mathcal{R}_{n_e,\alpha}$, decouples in two independent $(n_e/2)$-sheeted surfaces characterised by the parity of the sheets and therefore $\mathcal{E}(\alpha)(LT \ll 1) = 2\log\text{Tr}(\rho_{A_1}^{1/2}e^{i\hat{Q}_{A_1}\alpha})$. Conversely, this decoupling of the sheets does not occur at finite temperature. The lack of decoupling is manifested in the presence of the linear terms $\ell/\beta$ and $L/\beta$ in Eq. (127) which cannot be derived through a simple conformal mapping.

We also present the high-temperature limit of the charged probability in Eq. (119), that is

$$\log N_1(\alpha) \simeq \begin{cases} -\frac{2\alpha^2}{\pi^2}\Big[\log\Big|\frac{\beta}{\pi\epsilon_N}\sinh(\frac{2\pi\ell_1}{\beta})\Big| - \frac{\pi\ell_1}{\beta}\Big] - \frac{\alpha^2 L}{2\pi\beta}, & |\alpha| \leq \frac{\pi}{2}, \\ -\frac{2(\alpha-\pi)^2}{\pi^2}\log\Big|\frac{\beta}{\pi\epsilon_N}\sinh(\frac{2\pi\ell_1}{\beta})\Big| - \frac{2\alpha\ell_1}{\beta} + \frac{2\alpha^2\ell_1}{\pi\beta} - \frac{\alpha^2 L}{2\pi\beta}, & |\alpha| > \frac{\pi}{2}. \end{cases} \tag{129}$$

## 5.2 A semi-infinite system

A simple generalisation of the previous calculation concerns the charged logarithmic negativity for a semi-infinite system. For free fermions, the semi-infinite geometry is obtained from the infinite one by cutting the interval $A_1$ in half. Because of the structure of the vertex operators correlations in Eq. (113), the entanglement in the semi-infinite system is equal to half of

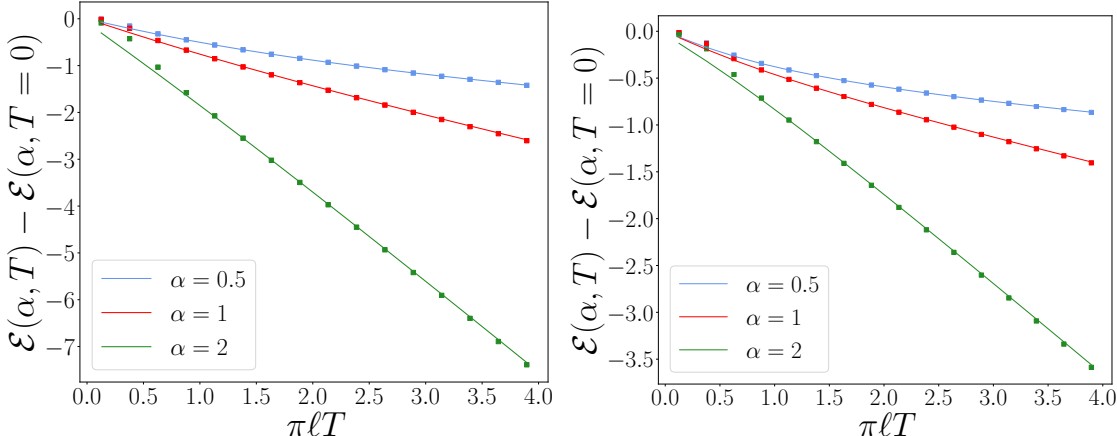

Figure 6: The charged logarithmic negativity for a bipartite geometry in the infinite-line (left) or semi-infinite (right). We set $L = 200$. Analytical prediction in Eqs. (127) and (132).

that of the infinite one in Eq. (116). Therefore, the charged logarithmic negativity of a finite interval with length $\ell_1 = r_1 L$ is given by

$$\log N_n^{(\nu)}(\alpha) = \log R_n - \frac{\alpha^2}{\pi^2 n} \log \left| \frac{\theta_1(2r_1|\tau)}{(\frac{\epsilon}{L}\partial_z \theta_1(0|\tau))} \right| \tag{130}$$

$$+ \sum_{k=-(n-1)/2}^{(n-1)/2} \log \left| \frac{\theta_\nu((\frac{k}{n} + \frac{\alpha}{2\pi n})(1 - 4r_1) + \frac{\varphi_n}{\pi} r_1 | \tau)}{\theta_\nu(0|\tau)} \right|. \tag{131}$$

The low and high temperature limits of this expression are obtained in analogy with the infinite line case, e.g. as $\tau \to 0$ we get

$$\mathcal{E}(\alpha) = \left(\frac{1}{4} - \frac{\alpha^2}{\pi^2}\right)\left[\log \left| \frac{\beta}{\pi \epsilon} \sinh(\frac{2\pi \ell_1}{\beta}) \right| - \frac{2\pi \ell_1}{\beta}\right] - \frac{\alpha^2 L}{4\pi \beta} + O(e^{-\pi LT}). \tag{132}$$

The correctness of these CFT charged negativities is tested against lattice calculations in Fig. 6. Here we plot $\mathcal{E}(\alpha, T) - \mathcal{E}(\alpha, T = 0)$ that turns out to be a universal function of $\pi \ell T$ and $\pi LT$, in agreement with Eqs. (127) and (132). As for the tripartite case, in order to test our final field theoretical results, we have taken into account the explicit expression for the cutoff $\epsilon$. Since it does not depend on the temperature, it can be extracted from the knowledge of the lattice charged moments $\text{Tr}(\rho_{A_1}^{1/2} e^{i\hat{Q}_{A_1}\alpha})$ at $T = 0$, derived in [96] and explicitly reported in the Appendix, see Eq. (182).

## 5.3 Symmetry resolution

As usual, the Fourier transform (75) is performed in the scaling regime with a saddle point approximation. Hence, the charged moments (72) can be truncated at Gaussian level in $\alpha$ as

$$N^{(\nu)}(\alpha)_n = R_n^{(\nu)} e^{-\frac{\alpha^2}{2} b_n}, \tag{133}$$

where

$$b_n = \frac{4}{\pi^2 n} \log \left| \mathcal{A}_n^{(1)} \frac{L\theta_1(r_1|\tau)}{\epsilon \partial_z \theta_1(0|\tau)} \right|, \tag{134}$$

and we used a quadratic approximation for

$$\log N_{n,1}^{(v)}(\alpha) = \mathcal{A}_n^{(0)} - \frac{2\alpha^2}{\pi^2 n} \log \mathcal{A}_n^{(1)}. \tag{135}$$

The RN in the sector $q$ are

$$R_n^{(v)}(q) = \frac{\mathcal{Z}_{R_1,n}(q)}{[p(q)]^n} = R_n^{(v)} \frac{\int_{-\pi}^{\pi} \frac{d\alpha}{2\pi} e^{-iq\alpha} e^{-\alpha^2 b_n/2}}{\left[ \int_{-\pi}^{\pi} \frac{d\alpha}{2\pi} e^{-iq\alpha} N_1(\alpha) \right]^n}, \tag{136}$$

and, through the saddle point approximation,

$$R_n^{(v)} \simeq R_n^{(v)} \sqrt{\frac{(2\pi b_N)^n}{2\pi b_n}} e^{-\frac{q^2}{2}\left(\frac{1}{b_n} - \frac{n}{b_N}\right)}, \qquad \mathcal{N}^{(v)}(q) \simeq \frac{1}{2}\left( e^{\mathcal{E}^{(v)}} \sqrt{\frac{2\pi b_N}{2\pi b}} e^{-\frac{q^2}{2}\left(\frac{1}{b} - \frac{1}{b_N}\right)} - 1 \right), \tag{137}$$

with

$$b = \lim_{n_e \to 1} b_{n_e}, \qquad b_N = \frac{4}{\pi^2} \log \left| \mathcal{A}_N^{(1)} \frac{L\theta_1(r_1|\tau)}{\epsilon_N \partial_z \theta_1(0|\tau)} \right|. \tag{138}$$

The explicit expressions for the cutoff $\epsilon$ and $\epsilon_N$ can be found in Eqs. (182) and (183a), respectively. When the $O(1)$ terms are negligible with respect to the leading order ones, $b \simeq b_N$ and we find the exact equipartition of negativity in the different charge sectors at leading order, i.e. $\mathcal{N}^{(v)}(q) \simeq \mathcal{N}^{(v)}$, on the same lines as for the tripartite case. We used that the leading contribution to the integral $p(q)$ comes from the region near the saddle point $\alpha = 0$, despite the presence of two local maxima at $\alpha = \pm\pi$. This is possible as long as $T > 0$: when $T \to 0$, the secondary maxima become degenerate with the one in 0 and they cannot be neglected. Their degeneracy is indeed related to the fact that $p(q)$ becomes zero for all odd $q$.

Finally, it is worth reporting the high temperature limits of the variances in Eq. (138), that following the steps in Section 5.1, simplify as

$$\begin{aligned} b &= \lim_{n_e \to 1} \frac{4}{\pi^2 n_e} \left[ \log \left| \frac{\beta}{\pi\epsilon} \sinh\left(\frac{\pi\ell_1}{\beta}\right) \right| - \frac{\pi\ell_1}{\beta} + \frac{\pi L}{4\beta} \right], \\ b_N &= \frac{4}{\pi^2} \left[ \log \left| \frac{\beta}{\pi\epsilon_N} \sinh\left(\frac{\pi\ell_1}{\beta}\right) \right| - \frac{\pi\ell_1}{\beta} + \frac{\pi L}{4\beta} \right]. \end{aligned} \tag{139}$$

The symmetry resolved negativities for a single interval embedded inside a semi-infinite chain or in the low-temperature regime are straightforwardly derived with minor modifications of the above calculations.

Also for this bipartite geometry it is instructive to identify the first term breaking equipartition. Expanding to order $O((\log L)^{-2})$ the above expressions, we get

$$\mathcal{N}^{(v)}(q) \simeq \mathcal{N}^{(v)} \left( 1 - \frac{\tilde{\gamma}}{\log|\theta_1(r_1|\tau)(\partial_z \theta_1(0|\tau)/L)|)} - \frac{q^2 \tilde{\gamma}\pi^2}{4(\log|\theta_1(r_1|\tau)(\partial_z \theta_1(0|\tau)/L)^{-1}|)^2} \right), \tag{140}$$

where $\tilde{\gamma} = \frac{\pi^2}{8} \log(\epsilon_N \mathcal{A}/(\epsilon \mathcal{A}_N))$. Since $\mathcal{A}/\mathcal{A}_N \simeq 1$, $\tilde{\gamma}$ can be explicitly computed through the results for $\epsilon$ and $\epsilon_N$ found in Appendix C.

Our analytic results for the symmetry resolved negativity are compared with the numerical data in the left panel of Figure 7. The equipartition of negativity is broken for all the considered values of $\ell, T, L$ and the effect is more evident as $|q|$ is increased. However, Eq. (137) can capture the smooth part of these corrections to the scaling, as shown by the full line in the plots. As $\ell \gg 1/T$, the corrections due to the presence of the maxima at $\alpha = \pm\pi$ in $N_1(\alpha)$

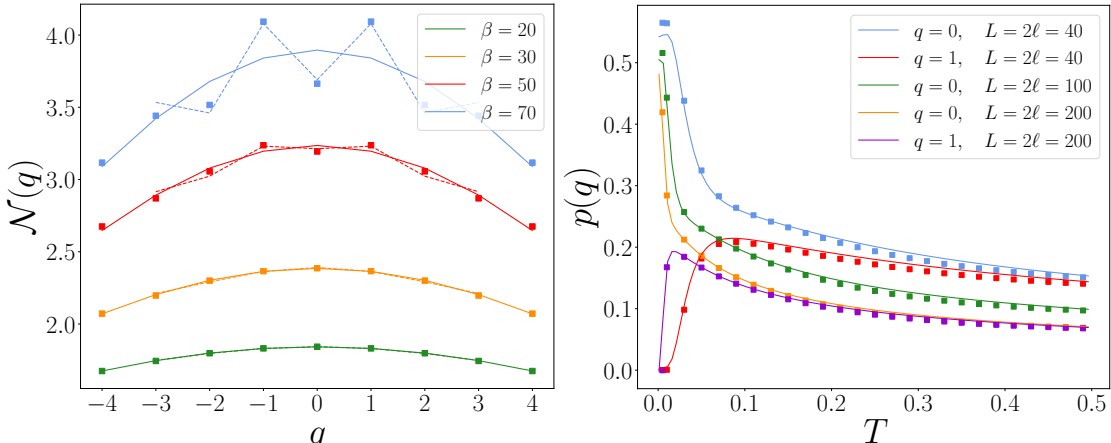

Figure 7: Left panel: The imbalance-resolved negativity as a function of $q$ in a bipartite geometry. The subsystem size is fixed, $\ell_1 = 40$, the total system sizes is also fixed, $L = 200$, while $T$ is varied. The coloured full lines represent Eq. (137) while the dashed ones represent Eq. (136) in the replica limit. Right panel: The probability of finding $q$ as outcome of a measurement of $\hat{Q}$. As the state becomes pure (i.e. $T \to 0$), $p(q) \to 0$ for odd $q$, as explained in Sec. 2.3. The full lines correspond to the Fourier transforms of the charged probability in (119) without saddle-point approximation.

become smaller and the imbalance resolved negativity flattens in $q$, mainly as a consequence of the lowering of the maxima at $\alpha = \pm\pi$ in $N_1(\alpha)$, see Fig. 4. We also check the correctness of our prediction for $p(q)$ in the right-panel of the same figure: we can observe that as the state becomes pure (i.e. $T \to 0$), $p(q) \to 0$ for odd $q$, as explained above. As already stressed many times, the divergent behaviour of negativity in the same charge sector is a consequence of the fact that the imbalance is no longer the right quantum number to resolve the entanglement.

# 6 Conclusions

We studied the entanglement negativity in systems with a conserved local charge and we found it to be decomposable into symmetry sectors. The partial TR operation does not spoil the result found for the standard partial transposition operation [91]: the resulting operator that commutes with the partial TR density matrix is not the total charge, but rather an imbalance operator, which is essentially the difference operator between the charge in the two regions. We introduced a normalised version of the charge imbalance resolved negativity (both fermionic and bosonic) which has the great advantage to be an entanglement proxy also for the symmetry sectors, e.g. it vanishes if the standard partial transpose has only positive eigenvalues in the sector. The price to pay is that the normalised symmetry resolved negativity diverges (for some sectors) in the limit of pure states, as a consequence of the fact that the imbalance is no longer the best quantity to resolve the entanglement. Another interesting property of this normalisation for the sector partial transpose is the *negativity equipartition*, i.e. the entanglement is the same in all imbalance sectors, in full analogy to entropy equipartition for pure states [93].

We then considered the (1+1)d CFT corresponding to free massless Dirac fermions at finite temperature $T$ and finite size $L$. We derived field theory predictions for the distribution of negativity in both tripartite and bipartite settings (i.e. the entanglement between two adjacent intervals and the one between one interval and the remainder, respectively). We tested our

prediction against numerical computations for a lattice version of the Dirac field, in which the non-universal terms are fixed from exact analytic computations. In both geometries, we find that, at leading order, the charge imbalance resolved negativity satisfies entanglement equipartition. We identify the subleading terms responsible for the breaking of equipartition in the lattice model.

There are different aspects that our manuscript leaves open for further study. The first one concerns the calculation of the time evolution of the charged and imbalance resolved negativity to understand if and how the quasiparticle picture remains true within the sectors of an internal local symmetry of a quantum many-body system, as recently done for the resolved entropies [95]. Secondly, one may use the corner transfer matrix to investigate the symmetry decomposition of negativity in gapped one-dimensional models by combining former studies of the total negativity [42] with those for symmetry resolution [99]. Eventually, the generalisation of one-dimensional results to higher dimensions can be done using the dimensional reduction approach, as already done for the total negativity in [85]. Decoupling the initial d-dimensional problem into one-dimensional ones in a mixed space-momentum representation [100] would allow to generalise the above results to higher dimensional Fermi surfaces.

## Acknowledgments

We thank Marcello Dalmonte, Giuseppe Di Giulio, Moshe Goldstein, and Vittorio Vitale for useful discussions. The authors acknowledge support from ERC under Consolidator grant number 771536 (NEMO).

## Appendix A   Numerical methods

In this first appendix, we report how to numerically calculate the charged negativity associated with the partial TR (15) for free fermions on a lattice described by the hopping Hamiltonian on a chain

$$H_{FF} = -\sum_{j=0}^{L-2} f_{j+1}^\dagger f_j + f_0^\dagger f_{L-1} + \text{H.c.}, \tag{141}$$

with anti-periodic condition (corresponding to the $\nu = 3$ sector discussed in the main text). The technique is a straightforward generalisation to $\alpha \neq 0$ of the one presented in [85]. This method is used throughout the main text to obtain all lattice numerical results.

Even though we use the computational basis of the Majorana modes, for particle-number conserving systems such as the lattice model in Eq. (141), the covariance matrix is simplified into the form $\sigma_2 \otimes \Gamma$, with $\Gamma = \mathbb{I} - 2C$, $C_{ij} = \text{Tr}(\rho f_i^\dagger f_j)$ is the correlation matrix and $\sigma_2$ is the second Pauli matrix (see Ref. [86] for a more detailed discussion). For a thermal state, the single-particle correlator reads

$$C_{ij} = \sum_k \frac{u_k^*(i) u_k(j)}{e^{\beta \omega_k} + 1}, \tag{142}$$

where $\omega_k$ and $u_k(i)$ are the single-particle eigenvalues and eigenvectors of the Hamiltonian (141). For a bipartite Hilbert space $\mathcal{H}_A \otimes \mathcal{H}_B$ where $A = A_1 \cup A_2$, the covariance matrix takes a block form

$$\Gamma = \begin{pmatrix} \Gamma_{11} & \Gamma_{12} \\ \Gamma_{21} & \Gamma_{22} \end{pmatrix}, \tag{143}$$

where $\Gamma_{11}$ and $\Gamma_{22}$ are the reduced covariance matrices of the two subsystems $A_1$ and $A_2$, respectively, while $\Gamma_{12}$ and $\Gamma_{21}^\dagger$ contain the cross correlations between them. By simple Gaussian

states' manipulations, the correlation matrices associated with $\rho_A^{R_1}$, $(\rho_A^{R_1})^\dagger$ can be written as [85]

$$\Gamma_\pm = \begin{pmatrix} -\Gamma_{11} & \pm i\Gamma_{12} \\ \pm i\Gamma_{21} & \Gamma_{22} \end{pmatrix}. \tag{144}$$

The objects we are interested in are $N_{n_e} = \text{Tr}[(\rho^{R_1}(\rho^{R_1})^\dagger)^{n_e/2} e^{i\hat{Q}\alpha}]$ and $N_1(\alpha) = \text{Tr}[\rho^{R_1} e^{i\hat{Q}\alpha}]$. The imbalance of the relativistic Dirac field corresponds to the discretised operator $\hat{Q} = \hat{N}_{A_1 \cup A_2} - 1/2(\ell_1 + \ell_2)$. Notice that in this basis, $\hat{Q}$ is not the difference, but the sum of the number operators. Furthermore, it presents a shift compared to the number operator of the non-relativistic fermions. The single particle correlation matrix associated to the normalised composite density operator $\rho_x = \rho^{R_1}(\rho^{R_1})^\dagger / Z_x$ is [85, 126]

$$\Gamma_x = (\mathbb{1} + \Gamma_+ \Gamma_-)^{-1}(\Gamma_+ + \Gamma_-), \tag{145}$$

where the normalisation factor is $Z_x = \text{Tr}(\Gamma_x) = \text{Tr}(\rho_A^2)$. In terms of eigenvalues of correlation matrices, we can write [85]

$$\log N_n(\alpha) = -i\alpha \frac{\ell_1 + \ell_2}{2} + \sum_{j=1}^N \log\left[\left(\frac{1 - v_j^x}{2}\right)^{n/2} + e^{i\alpha}\left(\frac{1 + v_j^x}{2}\right)^{n/2}\right] + \frac{n}{2}\sum_{j=1}^N \log\left[\zeta_j^2 + (1 - \zeta_j)^2\right], \tag{146}$$

where $v_j^x$ and $\zeta_j$ are eigenvalues of the matrices $\Gamma_x$ (145) and $C$ (142), respectively. In terms of the eigenvalues $v$'s of $\Gamma_\pm$ (144) ($\Gamma_+$ and $\Gamma_-$ have the same spectrum), the charged normalisation $N_1(\alpha)$ is

$$\log N_1(\alpha) = -i\alpha \frac{\ell_1 + \ell_2}{2} + \sum_{j=1}^N \log\left[\left(\frac{1 - v_j}{2}\right) + e^{i\alpha}\left(\frac{1 + v_j}{2}\right)\right]. \tag{147}$$

Taking the Fourier transform of the numerical data for $N_{n_e}(\alpha)$ and $N_1(\alpha)$, we finally obtain the imbalance resolved negativities.

# Appendix B  Mode expansion of charged moments of $\rho_A$ and $\rho_A^{R_1}$

Following Ref. [123], we report some details about the transformation of the trace formulas into a product of $n$ decoupled partition functions for non-interacting systems with conserved $U(1)$ charge. As mentioned in the main text, after diagonalising the twist matrices a partition function on a multi-sheet geometry can be decomposed as

$$Z_n(\alpha) = \prod_k Z_{k,n}(\alpha), \qquad Z_{k,n}(\alpha) = \langle e^{i\int d^2x A_\mu^k j_k^\mu}\rangle, \tag{148}$$

in which

$$\epsilon^{\mu\nu}\partial_\nu A_{k,\mu}(x) = 2\pi\sum_{i=1}^{2p} v_{k,i}(\alpha)\delta(x - u_i), \tag{149}$$

where $2\pi v_{k,i}(\alpha)$ is the vorticity of gauge flux determined by the eigenvalues of the twist matrix, $p$ are the intervals defined between a pair of points $u_{2i-1}$ and $u_{2i}$. The vorticities satisfy

the neutrality condition $\sum_i \nu_{k,i}(\alpha) = 0$ for every $k$. As already stressed, there are several representations of the partition function $Z_{k,n}(\alpha)$. In order to obtain the asymptotic behaviour, one needs to take the sum over all the representations

$$\tilde{Z}_{k,n}(\alpha) = \sum_{\{m_i\}} Z_{k,n}^{(m)}(\alpha), \quad Z_{k,n}^{(m)}(\alpha) = \langle e^{i \int d^2 x A_{k,\mu}^{(m)} j_k^\mu} \rangle \, \epsilon^{\mu\nu} \partial_\nu A_{k,\mu}^{(m)}(x) = 2\pi \sum_{i=1}^{2p} \tilde{\nu}_{k,i}(\alpha) \delta(x - u_i),$$

(150)

where $m_i$ is a set of integers and $\tilde{\nu}_{k,i}(\alpha) = \nu_{k,i}(\alpha) + m_i$ are shifted flux vorticities obeying $\sum_i m_i = 0$ because of neutrality condition. By the bosonisation technique, we may write

$$\tilde{Z}_{k,n}(\alpha) = E_{\{m_i\}} \prod_{i<j} \frac{1}{|u_i - u_j|^{-2\tilde{\nu}_{k,i}(\alpha)\tilde{\nu}_{k,j}(\alpha)}} \xrightarrow{\ell \to \infty} \tilde{Z}_{k,n}(\alpha) \sim \sum_{\{m_i\}} \frac{E_{\{m_i\}}}{\ell^{\sum_i \tilde{\nu}_{k,i}^2(\alpha)}},$$

(151)

where $\ell$ is a length scale and we absorbed all non-universal effects (e.g. cutoff and microscopic details in the case of lattice models) in the constants $E_{\{m_i\}}$. In the large $\ell$ limit, the leading order term(s) is (are) the one(s) which minimises the quantity $\sum_i \tilde{\nu}_{k,i}^2(\alpha)$. As shown in the following appendix, this is identical to (and consistent with) the condition derived from the generalised Fisher-Hartwig conjecture. We now carry out this procedure for $N_{n_e}(\alpha)$ in (74) for two adjacent intervals. We need to minimise the quantity

$$f_{m_1 m_2 m_3}(\nu) = (\nu - 1/2 + m_1)^2 + (\nu + m_2)^2 + (-2\nu + 1/2 + m_3)^2,$$

(152)

for a given $\nu = \frac{k}{n} + \frac{\alpha}{2\pi n}$, with $k = -\frac{n-1}{2}, \dots, \frac{n-1}{2}$, by finding the integers $(m_1, m_2, m_3)$ constrained by $\sum_i m_i = 0$. The triplet $(m_1, m_2, m_3)$ that minimises Eq. (152) is

$$\begin{cases} (0,0,0), & \nu \geq 0, \\ (1,0,-1), & \nu < 0. \end{cases}$$

(153)

For the charged probability in Eq. (87), we need to minimise

$$f_{m_1 m_2 m_3}(\nu) = (\nu + m_1)^2 + (\nu + m_2)^2 + (-2\nu + m_3)^2,$$

(154)

for $\nu = \frac{\alpha}{2\pi}$. In this case, the minimising sets of triplets are

$$\begin{cases} (0,0,0), & |\nu| \leq 1/3, \\ (-1,0,1),(0,-1,1), & \nu > 1/3, \\ (1,0,-1),(0,1,-1), & \nu < -1/3, \end{cases}$$

(155)

where in the two last lines the two reported triplets are degenerate.

A similar derivation can be carried out for $N_{n_e}(\alpha)$ in Eq. (74) for the geometry in Fig. 5. The quantity to minimise is

$$f_{m_1 m_2 m_3 m_4}(\nu) = (\nu + m_1)^2 + (-\nu + m_2)^2 + (-2\nu + 1/2 + m_3)^2 + (2\nu - 1/2 + m_4)^2,$$

(156)

where now we have four integers $(m_1, m_2, m_3, m_4)$ constrained by $\sum_i m_i = 0$. The quadruplet $(m_1, m_2, m_3, m_4)$ that minimises Eq. (158) is

$$\begin{cases} (0,0,0,0), & \nu \geq 0, \\ (0,0,1,-1), & \nu < 0. \end{cases}$$

(157)

For the charged probability in Eq. (119), where $\nu = \frac{\alpha}{2\pi}$, we have to minimise

$$f_{m_1 m_2 m_3 m_4}(\nu) = (\nu + m_1)^2 + (-\nu + m_2)^2 + (-2\nu + m_3)^2 + (2\nu + m_4)^2,$$

(158)

by choosing the quadruplet

$$\begin{cases} (0,0,0,0), & |\alpha| \leq \pi/2, \\ (0,0,1,-1), & |\alpha| > \pi/2. \end{cases}$$

(159)

# Appendix C  Lattice-dependent terms and Fisher-Hartwig conjecture

We focus on the evaluation of the explicit cutoffs induced by the lattice for the charged Rényi negativity, for the ground state of the Hamiltonian (141). Analogous results for the charged Rényi and entanglement entropies have already been worked out in [96]. The evaluation of the charged negativity relies on the Fisher-Hartwig conjecture for the determinant of Toeplitz matrices. Here we closely follow the derivation for the negativity at $T = 0$ [81]. We focus on a chain of length $L$ with sites labelled by $j = [0, L-1]$; the intervals are $A_1 = [0, \ell - 1]$ and $A_2 = [\ell_1, \ell_2 - 1]$. The non-universal additive constant does not depend on the finiteness of the chain. Denoting by $\tilde{u}_i(r_j) = \langle r_j | \tilde{u}_i \rangle$ the single particle eigenstate, the wave function describing the ground state of the Hamiltonian has the form of a Slater determinant

$$\langle \{r_j\} | \Psi \rangle = \det[\tilde{u}_i(r_j)]. \tag{160}$$

Hence, the partition function $Z_{R_1,k}(\alpha)$ in Eq. (74) is

$$Z_{R_1,k}(\alpha) = \det M^{R_1,k}_{mm'}(\alpha) = \det \left( \langle \tilde{u}_m | T^{R_1}_{\alpha,k} | \tilde{u}_{m'} \rangle \right), \tag{161}$$

where $T^{R_1}_\alpha$ is a diagonal matrix whose entries $[T^{R_1}_\alpha]_{jj} = T^{R_1}_{\alpha,k}(j)$ are

$$T^{R_1}_{\alpha,k}(j) = \begin{cases} e^{i\varphi - 2\pi i \frac{k}{n} - i \frac{\alpha}{n}} & 0 \le j < \ell_1, \\ e^{i 2\pi \frac{k}{n} + i \frac{\alpha}{n}} & \ell_1 \le j < \ell_1 + \ell_2, \\ 1 & \ell_1 + \ell_2 \le j < L, \end{cases} \tag{162}$$

where the phase are fixed according to the conventions in Figure 1. Writing explicitly the single particle eigenstates as plane waves $\tilde{u}_i(r_j) = \frac{1}{\sqrt{L}} e^{i \frac{\pi m}{L} j}$, $m = \pm 1, \pm 3, ..., \pm(L/2-1)$, we obtain

$$M^{R_1,k}_{mm'}(\alpha) = \langle \tilde{u}_m | T^{R_1}_{\alpha,k} | \tilde{u}_{m'} \rangle = \frac{1}{L} \sum_{j=0}^{L-1} e^{-i \frac{\pi j}{L}(m-m')} T^{R_1}_{\alpha,k}(j) = \frac{1}{2\pi} \int_0^{2\pi} d\theta e^{-i\theta(m-m')/2} \tilde{T}^{R_1}_{\alpha,k}(\theta), \tag{163}$$

The last identity in (163) is obtained in the scaling regime $L \to \infty$, $\ell \to \infty$ and $\ell/L$ fixed and $\tilde{T}^{R_1}_{\alpha,k}(\theta)$ is the continuum limit ($j \to \frac{L\theta}{2\pi}$) of Eq. (162), i.e.

$$\tilde{T}^{R_1}_{\alpha,k}(\theta) = \begin{cases} e^{i\varphi - 2\pi i \frac{k}{n} - i \frac{\alpha}{n}} & 0 < \theta < \pi r_1 \\ e^{i 2\pi \frac{k}{n} + i \frac{\alpha}{n}} & \pi r_1 < \theta < \pi(r_1 + r_2), \\ 1 & \pi(r_1 + r_2) < \theta < 2\pi \end{cases} \tag{164}$$

where we introduced $r_i = 2\ell_i/L$. Hence, in this basis, the matrix $M^{R_1,k}_{mm'}(\alpha)$ is a Toeplitz matrix where $(m - m')/2 = 0, 1, 2, \dots, (L/2 - 1)$ which implies that the size of the matrix is $\frac{L}{2} \times \frac{L}{2}$.

The asymptotic evaluation of the determinant of a Toeplitz matrix is based on a special standard structure that we now describe. In general, a Toeplitz matrix has the form $T_L[\phi] = (\phi_{i-j})$, where $\phi_k$ is the $k$-th Fourier coefficient of the *symbol* $\phi(\theta)$. The Fisher-Hartwig conjecture gives the asymptotic behaviour of the determinant of Toeplitz matrices whose symbol admits a canonical factorisation as

$$\phi(\theta) = \psi(\theta) \prod_{r=1}^{R} t_{\beta_r, \theta_r}(\theta) u_{\alpha_r, \theta_r}(\theta), \tag{165}$$

where

$$t_{\beta_r,\theta_r}(\theta) = \exp[-i\beta_r(\pi - \theta + \theta_r)], \quad \theta_r < \theta < 2\pi + \theta_r, \tag{166}$$

$$u_{\alpha_r,\theta_r}(\theta) = (2 - 2\cos(\theta - \theta_r))^{\alpha_r}, \quad \mathrm{Re}[\alpha_r] > -\frac{1}{2}, \tag{167}$$

$\psi(\theta)$ is a smooth vanishing function with zero winding number and $R$ is the number of discontinuities of $\phi(\theta)$. For $L \to \infty$, the Fisher-Hartwig formula gives

$$\det T_L[\phi] = (\mathcal{F}[\psi])^L \left( \prod_{r=1}^{R} L^{\alpha_i^2 - \beta_i^2} \right) E_{\mathrm{FH}}, \tag{168}$$

where

$$\mathcal{F}[\psi] = \exp\left( \frac{1}{2\pi} \int_0^{2\pi} \ln \psi(\theta) d\theta \right), \tag{169}$$

and assuming that $\psi(\theta)$ admits the Wiener-Hopf factorisation

$$\psi(\theta) = \mathcal{F}[\psi]\psi_+(\exp(i\theta))\psi_-(\exp(-i\theta)). \tag{170}$$

Here $E[\psi] = \exp\left(\sum_{k=1}^{\infty} k s_k s_{-k}\right)$, with $s_k$ corresponding to the $k$-th Fourier coefficient of $\ln \psi(\theta)$, and $G$ are the Barnes $G$-function

$$G(1+z) = (2\pi)^{z/2} e^{-(z+1)z/2 - \gamma_E z^2/2} \prod_{n=1}^{\infty} [(1 + z/n)^n e^{-z + z^2/(2n)}], \tag{171}$$

and $\gamma_E$ the Euler constant.

In the case of the negativity for two adjacent intervals and even $n = n_e$, the *symbol* $\phi(\theta)$ is given by

$$\phi(\theta) = \begin{cases} e^{i\pi - 2\pi i(k/n + \alpha/(2\pi n))}, & -\pi r_1 < \theta < 0, \\ e^{2\pi i(k/n + \alpha/(2\pi n))}, & 0 < \theta < \pi r_2, \\ 1, & \pi r_2 < \theta < 2\pi - \pi r_1. \end{cases} \tag{172}$$

Therefore, it has three discontinuities and admits the following canonical factorization:

$$\phi(\theta) = \psi(\theta) t_{\beta_1(k), -\pi r_1}(\theta) t_{\beta_2(k), \pi r_2}(\theta) t_{\beta_3(k), 0}(\theta), \tag{173}$$

where

$$\psi(\theta) = e^{i\pi r_1/2 - i\pi(\frac{k}{n} + \frac{\alpha}{2\pi n})(r_1 - r_2)}, \tag{174}$$

$$\beta(k) = \frac{k}{n} + \frac{\alpha}{2\pi n} = \beta_1(k) + \frac{1}{2} = \beta_2(k), \tag{175}$$

$$\beta_3(k) = -2\beta(k) + \frac{1}{2}, \tag{176}$$

so we can apply the conjecture with $R = 3$ and $\alpha_i = 0$. When $\beta < 0$, we have $|\beta_3| > 1/2$ and the FH conjecture in its original form breaks down (we can also use the most general hypothesis in which the Fisher-Hartwig conjecture works, i.e. if we introduce the seminorm $|||\beta||| = \max_{j,k} |\beta_k - \beta_k|$, where $1 \le j, k \le 3$, the conjecture has been verified for $|||\beta||| < 1$ [127]). In this case we should use the generalised Fisher-Hartwig conjecture, see e.g. [127, 128] in which one sums over all the inequivalent representations of the symbol, i.e. summing over all possible $\{\hat{\beta}_i\}$ such that

$$\hat{\beta}_i = \beta_i + n_i, \qquad \sum_{i=1}^{R} n_i = 0. \tag{177}$$

The leading terms is given by the set(s) of integers $\{n_i\}$ that minimises the function $\sum_{i=1}^{R} \hat{\beta}_i^2$. This is identical to the condition derived in Sec. B. We denote the corresponding set of solutions by $\mathcal{M}_\beta$. The Toeplitz determinant is sum of the standard form (168) corresponding to these solutions. Then, the asymptotic behaviour of the Toeplitz determinant in (161) is

$$
\begin{aligned}
Z_{R_1,k}(\alpha) = {}& (2 - 2\cos(2\pi\ell_1/L))^{-(|k/n+\alpha/(2\pi n)|-1/2)(2|k/n+\alpha/(2\pi n)|-1/2)} \\
& \times (2 - 2\cos(2\pi\ell_2/L))^{-|k/n+\alpha/(2\pi n)|(2|k/n+\alpha/(2\pi n)|-1/2)} \\
& \times (2 - 2\cos(2\pi(\ell_1+\ell_2)/L))^{|k/n+\alpha/(2\pi n)|(|k/n+\alpha/(2\pi n)|-1/2)} \\
& \times J_\alpha(k)(L)^{\Delta_k(\alpha)},
\end{aligned}
\tag{178}
$$

where $\Delta_k(\alpha)$ is given by Eq. (80) and $J_\alpha(k) = \prod_{i=1}^{3} G(1+\beta_i(k))G(1-\beta_i(k))$. Using standard manipulation of the Barnes G-function, we can rewrite the $O(1)$ terms as

$$
\begin{aligned}
\left(\frac{n^2-2}{4n} + \frac{3\alpha^2}{2\pi^2 n}\right)\log\epsilon = {}& \sum_{k=-\frac{(n-1)}{2}}^{k=\frac{(n-1)}{2}} \log 2^{2\Delta_k} J_\alpha(k) = -\log 2\left(\frac{n^2-2}{4n} + \frac{3\alpha^2}{2\pi^2 n}\right) \\
& -(1+\gamma_E)\left(\frac{n}{4} - \frac{1}{2n} + \frac{3\alpha^2}{2n\pi^2}\right) + \sum_{m=1}^{\infty} \frac{-2\pi^2 + n^2\pi^2 + 6\alpha^2}{4mn\pi^2} \\
& + \sum_{m=1}^{\infty} 2m\log \frac{(\frac{2}{m^3 n^3})^n \Gamma[\frac{2+n-2nm}{4} - \frac{\alpha}{2\pi}]\Gamma[\frac{2+n-2nm}{4} + \frac{\alpha}{2\pi}]\Gamma[\frac{1+n+2nm}{2} - \frac{\alpha}{2\pi}]\Gamma[\frac{1-n+2nm}{2} - \frac{\alpha}{2\pi}]}{\Gamma[\frac{2-n-2nm}{4} - \frac{\alpha}{2\pi}]\Gamma[\frac{2-n-2nm}{4} + \frac{\alpha}{2\pi}]\Gamma[\frac{1-n+2nm}{2} + \frac{\alpha}{2\pi}]\Gamma[\frac{1-n+2nm}{2} - \frac{\alpha}{2\pi}]}.
\end{aligned}
\tag{179}
$$

Eq. (179) is well approximated by the expansion at the second order in $\alpha$ since higher corrections $O(\alpha^4)$ are negligible for most practical purposes (this is in full analogy with the charged entropies [96]). In particular, in the replica limit $n_e \to 1$ we have

$$
-\left(\frac{1}{4} - \frac{3\alpha^2}{2\pi^2}\right)\log\epsilon \simeq 0.47295 - 0.29990\alpha^2.
\tag{180}
$$

In conclusion, the Fisher-Hartwig technique allows us to re-derive the leading CFT terms (at $T = 0$) and also provides the cutoff due to lattice regularisation.

Similar derivation can be carried out to compute the cutoff $\epsilon_N$ of the charged probability in Eq. (87) and we report the final result

$$
\frac{3\alpha^2}{2\pi^2}\log\epsilon_N \simeq -0.34514\alpha^2 \quad |\alpha| \le 2\pi/3
\tag{181a}
$$

$$
-\left(1 - \frac{|\alpha|}{\pi}\right)\frac{|\alpha|}{\pi}\log\epsilon_N \simeq -3.21853 + 1.93194|\alpha| - 0.30748\alpha^2 \quad |\alpha| > 2\pi/3.
\tag{181b}
$$

For a bipartite geometry, the lattice cutoff $\epsilon$ can be computed using the equivalence between the (charged) negativity and the (charged) $\frac{1}{2}$ Rényi entropy for pure states, exploiting the results of Ref. [96], i.e.

$$
-\left(\frac{1}{2} - \frac{2\alpha^2}{\pi^2}\right)\log\epsilon \simeq 0.94590 - 0.36978\alpha^2.
\tag{182}
$$

We also write down the final result for the cutoff $\epsilon_N$ of the charged probability in Eq. (119)

$$
\frac{2\alpha^2}{\pi^2}\log\epsilon_N \simeq -0.46020\alpha^2, \quad |\alpha| \le \pi/2,
\tag{183a}
$$

$$
\frac{(\pi-|\alpha|)^2}{\pi^2}\log\epsilon_N \simeq -0.46020(|\alpha| - \pi)^2, \quad |\alpha| > \pi/2.
\tag{183b}
$$

As already discussed in the main text, the knowledge of the exact expression for this non-universal quantities is relevant in order to test our analytical predictions against numerical data without any fitting parameter.

## Appendix D  Twisted partial transpose

For $T = 0$, the fermionic Rényi negativity in CFT (17) is equal [85] to the bosonic Rényi negativity for both even and odd values of $n$, also in the charged case [91]. The definition (17) has been employed since $\rho_A^{R_1}$ is not necessarily Hermitian. Then, the trace norm in terms of square root of the eigenvalues of the composite operator $\rho_x = \rho_A^{R_1}(\rho_A^{R_1})^\dagger$ provides a well-defined entanglement negativity for fermions. However, one can also introduce a hermitian partial transpose, which is suitable to define another fermionic negativity because of its real spectrum [86]. This is done by considering the composite operator $\tilde{\rho}_x = (\rho_A^{\tilde{R}_1})^2$, in terms of the twisted partial transpose $\rho_A^{\tilde{R}_1} = \rho_A^{R_1}(-1)^{F_1}$, with $(-1)^{F_1}$ the fermion number parity in $A_1$. The associated charged moments are $\tilde{N}_n = \text{Tr}[(\rho_A^{\tilde{R}_1})^n e^{i\hat{Q}\alpha}]$. The use of the composite operator $\tilde{\rho}_x$ is distinct from the previous one. In fact, the $T_\alpha^{\tilde{R}_1}$ matrix which glues together $\rho_A^{\tilde{R}_1}$ is

$$
T_\alpha^{\tilde{R}_1} = \begin{pmatrix} 0 & 0 & \cdots & -e^{-i\alpha/n} \\ e^{-i\alpha/n} & 0 & & \\ 0 & e^{-i\alpha/n} & & \ddots \\ & & \ddots & \ddots \end{pmatrix}.
\tag{184}
$$

The technical difference is that the twist phases of the two intervals are now $e^{2\pi i(\frac{k}{n} + \frac{\alpha}{2\pi n} - \frac{\varphi_n}{2\pi})}$ and $e^{-2\pi i(\frac{k}{n} + \frac{\alpha}{2\pi n})}$, with $\varphi_n = \pi$ for both $n$ even and odd. This means that for $n$ even the two Rényi negativity are equal (and indeed the negativity may be obtained from the replica limit $n_e \to 1$ of both). Hence, we report the final results for the two geometries studied in the main text for $n = n_o$.

 **1. Adjacent intervals:** The spin-independent part of the moments of negativity are given by

$$
\begin{aligned}
\log \tilde{N}_{n_o,0}(\alpha) = \log \tilde{R}_{n_o} &- \frac{\alpha^2}{2\pi^2 n_o} \log |\theta_1(r_1|\tau)^2 \theta_1(r_2|\tau)^2 \theta(r_1 + r_2|\tau)^{-1}(\tfrac{\epsilon}{L}\partial_z \theta_1(0|\tau))^{-3}| \\
&+ \frac{|\alpha|}{2n_o \pi} \log |\theta_1(r_1|\tau)^3 \theta_1(r_2|\tau) \theta(r_1 + r_2|\tau)^{-1}(\tfrac{\epsilon}{L}\partial_z \theta_1(0|\tau))^{-3}|,
\end{aligned}
\tag{185}
$$

while the spin structure dependent term is the same as Eq. (84) with $\varphi_{n_o} = \pi$.
 **2. Bipartite geometry:**  In this case, one has

$$
\begin{aligned}
\log \tilde{N}_{n_o,0}(\alpha) = \log \tilde{R}_{n_o} &- \frac{\alpha^2}{2\pi^2 n_o} \log \left| \frac{\theta_1(r_1|\tau)^4 \theta_1\left(\frac{r_2}{2}|\tau\right)^4 \theta_1(r_1 + r_2|\tau)}{\theta_1\left(r_1 + \frac{r_2}{2}|\tau\right)^4 (\tfrac{\epsilon}{L}\partial_z \theta_1(0|\tau))^5} \right| \\
&+ \frac{|\alpha|}{\pi n_o} \log \left| \frac{\theta_1(r_1|\tau)^2 \theta_1\left(\frac{r_2}{2}|\tau\right)}{\theta_1\left(r_1 + \frac{r_2}{2}|\tau\right)(\tfrac{\epsilon}{L}\partial_z \theta_1(0|\tau))^2} \right|,
\end{aligned}
\tag{186}
$$

for the spin-independent part while the spin structure dependent term is the same as Eq. (117) with $\varphi_{n_o} = \pi$.

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
