# Peer review of "Symmetry decomposition of negativity of massless free fermions"

_SciPost Physics, doi:SciPost Phys. 10, 111 (2021)_

## Round 1 · Referee Report · Anonymous (Referee 1) · 2021-4-5

Strengths

1- The paper addresses a hot topic in condensed matter physics. 2- The results are new and original. 3- The paper is written in a clear and pedagogical way. 4- The paper correctly cites the relevant literature.

Weaknesses

None

Report

This paper discusses various measures of entanglement for free-fermionic models in the presence of a conserved charge. It focuses on the so-called fermionic negativity, a measure of entanglement defined from the trace of the reduced density matrix wherein a time-reversal is applied to the degrees of freedom of one of the subsystems, and on its resolution between different symmetry sectors. The authors perform a careful and complete CFT analysis of this quantity for a free Dirac field. Their calculation uses the replica trick and the known correlation functions for vertex operators, and leads to non-trivial predictions for the scaling behaviour of the fermionic negativity and of its distribution between the symmetry sectors. These results are shown to be consistent with numeric experiments performed on a free-fermionic lattice model. The authors also discuss the asymptotic behaviour in the limits of low and high temperature, which will most probably be useful for comparisons with real experiments.

The paper is well-written. I have a very good impression of this paper overall. The first sections give a welcome pedagogical review of known results on this topic, with toy examples that help the reader and justify the more complex calculations of the later sections. The authors compare their results with the existing literature throughout. I strongly recommend the publication of this article in Scipost Phys.

Here is nevertheless a short list of more specific comments/questions/typos:

  • It would be interesting to have the authors comment on which of their results are expected to hold for models like the XXZ spin-chain which are fermionic but not free-fermionic.

  • I am not sure I understand why eq. (10) is claimed to be a partial time-reversal. Does it not simply represent the basis of the Hilbert space in terms of occupation numbers?

  • In eq. (29), the parameter epsilon is (up to that point) undefined. If it is identical to the cut-off parameter discussed later, I think it would be worthwhile to discuss its signification in section 2.2.1 already.

  • Likewise it seems that he parameter \bar q in (30) is undefined.

  • Also about (30), it would be useful to know the physical meaning of the saddle point approximation here, namely in what regimes of the parameters (β, L, …) it is expected to hold.

  • A factor of exp(i α/n) appears to be missing in (50) in the lower left corner of the matrix.

  • Presumably, the correct equation to be referenced above (86) is (77), not (112).

  • It would be useful if the boundary conditions in (140) used for the numerical analysis were specified.

  • validity: top
  • significance: top
  • originality: high
  • clarity: top
  • formatting: excellent
  • grammar: good

Author:  Sara Murciano  on 2021-04-20  [id 1370]

(in reply to Report 1 on 2021-04-05)

We thank the referee for the very positive report on our manuscript and for noting all typos/wrong expressions.
We took all suggestions into account, namely

R: It would be interesting to have the authors comment on which of their results are expected to hold for models like the XXZ spin-chain which are fermionic but not free-fermionic.
A: The results for the bosonic negativity collected in section 2.2.1 are valid for a system of interacting fermions (i.e. for a free compact boson with arbitrary compactification radius)
at zero temperature. Already generalising this result at finite temperature is very difficult and likely out of reach (it has not been done for the entanglement entropy of two intervals).
The technique used in this manuscript does not apply for the interacting case becuase the decoupling of the k-modes in the replica space does not hold.
We added a sentence specifying these difficulties

R: I am not sure I understand why eq. (10) is claimed to be a partial time-reversal. Does it not simply represent the basis of the Hilbert space in terms of occupation numbers?
A: Yes, it simply represents the basis of the Hilbert space in terms of occupation numbers, so we fixed it.

R: In eq. (29), the parameter epsilon is (up to that point) undefined. If it is identical to the cut-off parameter discussed later, I think it would be worthwhile to discuss its signification in section 2.2.1 already. Likewise, it seems that the parameter \bar q in (30) is undefined.
A: In eq. 29 the parameter epsilon is the cut-off parameter discussed later, we have specified it together with \bar{q} and the meaning of the saddle-point approximation.

R: Also about (30), it would be useful to know the physical meaning of the saddle point approximation here, namely in what regimes of the parameters (?, L, ?) it is expected to hold.

R: A factor of exp(i ?/n) appears to be missing in (50) in the lower left corner of the matrix.
Thanks, corrected.

R: Presumably, the correct equation to be referenced above (86) is (77), not (112).
A: Thanks, fixed.

R: It would be useful if the boundary conditions in (140) used for the numerical analysis were specified.
A: We specified that we are dealing with antiperiodic boundary conditions.

---

## Round 1 · Referee Report · Anonymous (Referee 2) · 2021-4-14

Strengths

1- Nontrivial new results on symmetry resolved aspects of entanglement measures. 2- Clearly written.

Weaknesses

1- Not totally clear what symmetry resolution brings, compared to regular entanglement measures.

Report

This paper studies the symmetry-resolved --with respect to charge imbalance-- negativity of a free Dirac field, mostly with path integral and bosonisation techniques. In particular, they derive new exact formulas in various bipartitions and tripartitions, and also at finie temperature.

Overall this is a good paper, it does contain some nontrivial results on a recently popular topic. The findings presented here will be useful to the community interested in entanglement scaling both in condensed matter and high energy physics. The calculations are clearly explained, which makes the paper a pleasant read.

For this reason I recommend publication.

I have a few minor remarks on the text, see below:

1) Page 3, last line. 'pointed out that also when' also can be removed.

2) Page 5, after (7). 'the reason of' $\to$ 'the reason for'.

3) Page 6, line 3. 'so to distinguish' $\to$ 'so as to distonguish'.

4) Page 7, section 2.2. 'contributions of the entanglement entropy' $\to$ 'contributions to the entanglement entropy'.

5) Page 8, top. It would be better to reshuffle the basis states in (20) so as to match the order of the terms in (21).

6) Page 8, after (22). 'We recall that this operator \hat{Q} is basis dependent'. I do not get the point: except for the identity, any operator is basis dependent. Also, 'the projector into' $\to$ 'the projector onto'.

7) Page 9, last paragraph. I find the whole discussion to be a little bit too long and convoluted.

8) Page 10, after (32). 'shown up only' $\to$ 'shown only'.

9) Page 13, before (48). 'the partition function on such modified' $\to$ 'the partition function of such a modified'.

10) Page 15, after (58). I do not understand what 'circuits' means in this context.

11) Page 17, after (71). 'entanglement equipartition holds also at finite size and temperature, at least for a free Diract field'. The authors should comment on what would happen for interacting fermions that map to a free bosonic theory with different compactification radius, where equipartition of the entanglement entropy is known at zero temperature.

12) Page 20, after (83). 'reminiscent of what' $\to$ 'reminiscent of what was'.

13) Appendix C. The notations at the beginning are confusing, in particular (161), (162). $L$ still appears on the rhs in (162), and the symbol should not depend on the matrix size to get a Toeplitz matrix. 'scaling regime $L\to\infty$, $\ell\to \infty$, $\ell/L$ fixed'. It is not clear to me what is exactly $L$, as it does not appear to be the chain size in (140). It would be desirable to explain more clearly what problem is being solved here.

  • validity: top
  • significance: high
  • originality: good
  • clarity: top
  • formatting: excellent
  • grammar: good

Author:  Sara Murciano  on 2021-04-20  [id 1371]

(in reply to Report 2 on 2021-04-14)

We thank the referee for considering the paper clearly written and useful to the community interested in entanglement scaling both in condensed matter and high energy physics. We also thank the referee for pointing out all the typos/wrong expressions that we modified.

6) R: Page 8, after (22). 'We recall that this operator \hat{Q} is basis dependent'. I do not get the point: except for the identity, any operator is basis dependent. A: We agree that all operator are basis dependent. Here we are referring to the fact that the functional form in terms of the number operar N_1 and N_2 is basis dependent. In the Fock basis, the imbalance is the difference of the number operators, as the name says. Instead in the Majorana basis in the appendix it is the sum of the numbers of Majoranas (up to a constant). We tried to clarify this point adding an explicit comment.

7) R: Page 9, last paragraph. I find the whole discussion to be a little bit too long and convoluted. A: We agree that is long and pedantic, but we believe that it is a very important issue for the experts and better to repeat ourselves than let the point lost.

10) R: Page 15, after (58). I do not understand what 'circuits' means in this context. A: The circuits refer to the paths around the left and right endpoints of the i-th interval defined after Eq. (55).

11) R: Page 17, after (71). 'entanglement equipartition holds also at finite size and temperature, at least for a free Diract field'. The authors should comment on what would happen for interacting fermions that map to a free bosonic theory with different compactification radius, where equipartition of the entanglement entropy is known at zero temperature. A: This is the same question raised by referee 1. We report the same answer here again. The results for the bosonic negativity collected in section 2.2.1 are valid for a system of interacting fermions (i.e. for a free compact boson with arbitrary compactification radius) at zero temperature. Already generalising this result at finite temperature is very difficult and likely out of reach (it has not been done for the entanglement entropy of two intervals). The technique used in this manuscript does not apply for the interacting case because the decoupling of the k-modes in the replica space does not hold. We added a sentence specifying these difficulties

13) R: Appendix C. The notations at the beginning are confusing, in particular (161), (162). L still appears on the rhs in (162), and the symbol should not depend on the matrix size to get a Toeplitz matrix... A: We apologise if the notations in this section were very confusing. We have been a bit sloppy, because everything was taken from Ref. [81]. Now we have been very precise with all the definitions, making clear what is the scaling limit and showing that the matrix elements do not depend on L, as they should.

All other points by the referee were either typos or stylistic changes that we took all into account, but it is not worth to rewrite here.

---

## Round 2 · Referee Report · Anonymous (Referee 1) · 2021-4-23

Report

I am happy with the changes made by the authors and recommend this article for publication.

---

## Round 2 · Referee Report · Anonymous (Referee 2) · 2021-5-5

Report

The authors have addressed all my concerns, I recommend publication.

---

## Round 2 · Author Response

We are submitting a revised version of the manuscript "Symmetry decomposition of negativity of massless free fermions".
We would like to thank the editors for their work and the referees for
their useful comments and suggestions.

---

## Round 2 · List of Changes

• We modified all typos/misleading/wrong expressions.
  • We commented on the validity of our results for interacting models.
  • We tried to clarify the basis dependence of the charge operator.
  • We clarified the confusing notations in Appendix C.

---

## Editorial Decision

published